# Post-capillary venules are the key locus for transcytosis-mediated brain delivery of therapeutic nanoparticles

Krzysztof Kucharz [1,5 ✉], Kasper Kristensen [2,5], Kasper Bendix Johnsen[2,5], Mette Aagaard Lund[2], Micael Lønstrup [1], Torben Moos[3], Thomas Lars Andresen[2] & Martin Johannes Lauritzen[1,4 ✉]

Effective treatments of neurodegenerative diseases require drugs to be actively transported across the blood-brain barrier (BBB). However, nanoparticle drug carriers explored for this purpose show negligible brain uptake, and the lack of basic understanding of nanoparticle-BBB interactions underlies many translational failures. Here, using two-photon microscopy in mice, we characterize the receptor-mediated transcytosis of nanoparticles at all steps of delivery to the brain in vivo. We show that transferrin receptor-targeted liposome nanoparticles are sequestered by the endothelium at capillaries and venules, but not at arterioles. The nanoparticles move unobstructed within endothelium, but transcytosis-mediated brain entry occurs mainly at post-capillary venules, and is negligible in capillaries. The vascular location of nanoparticle brain entry corresponds to the presence of perivascular space, which facilitates nanoparticle movement after transcytosis. Thus, post-capillary venules are the point-of-least resistance at the BBB, and compared to capillaries, provide a more feasible route for nanoparticle drug carriers into the brain.

[1] Department of Neuroscience, Faculty of Health Sciences, University of Copenhagen, Copenhagen N, Denmark. [2] Department of Health Technology, Technical University of Denmark, Kongens Lyngby, Denmark. [3] Department of Health Science and Technology, Aalborg University, Aalborg Ø, Denmark. [4] Department of Clinical Neurophysiology, Rigshospitalet, Glostrup, Denmark. [5] These authors contributed equally: Krzysztof Kucharz, Kasper Kristensen, Kasper Bendix Johnsen. ✉email: kucharz@sund.ku.dk; mlauritz@sund.ku.dk

The blood-brain barrier (BBB) is impermeable to most blood-borne substances, protecting the fragile brain environment from potentially harmful insults[1]. The para-cellular entry of molecules from the blood to the brain is barred by junctional complexes between adjoining brain endothelial cells (BEC)[2]. Diffusion of molecules across BECs is possible but restricted to low-molecular-weight hydrophobic compounds. Most of them, however, including therapeutics, show negligible brain uptake due to rapid outward transport by efflux pumps to the bloodstream[3,4]. Macromolecules, e.g., proteins, can enter the brain by vesicular transport, i.e., transcytosis, but this route is highly selective and actively suppressed by recently identified homeostatic mechanisms[5,6]. Consequently, the BBB precludes more than 99% of neuroprotective compounds from reaching the brain, rendering central nervous system (CNS) disorders resistant to most conventional therapies[3,7,8].

Drug delivery systems that aim to adapt receptor-mediated transcytosis (RMT) to shuttle therapeutic cargo across the BBB are currently at the forefront of modern therapeutic approaches against brain diseases[9–11]. The flagship ferrying receptor on BECs used for that purpose is the transferrin receptor (TfR)[8,12,13]. Neuroprotective compounds show enhanced brain delivery when coupled to TfR ligands, e.g., antibody fragments[14,15], but require chemical conjugation to the targeting moiety. In comparison, nanoparticle drug carriers are versatile delivery vehicles that can encapsulate large payloads of xenobiotics with a wide range of biophysical characteristics, offering diverse and unique therapeutic opportunities[9–11]. Liposome nanoparticles functionalized with TfR ligands represent a promising drug delivery approach tested currently in preclinical trials in brain cancer, stroke, and Parkinson's, Alzheimer's, and Huntington's disease, but the levels of nanoparticle transport into the brain need to improve to meet dosage requirements and reach clinical significance[12,13].

To improve drug delivery, it is crucial to understand how the BBB handles drug nanocarriers in the living brain, but conventional experimental techniques only provide limited mechanistic insights. Whole-brain imaging techniques, such as PET or MRI, are insufficient to resolve the spatio-temporal characteristics of single nanocarrier-BBB interactions in vivo. Consequently, the events between administration and detection of therapeutics in the brain are obscured. Our current knowledge is extrapolated from chemically processed tissue[3,13,16], which erases all information on the dynamic processes at the BBB. In addition, the vessel microanatomy differs between arterioles, capillaries, and venules[17,18]. This principal feature of the brain is overlooked in drug delivery studies, and how it impacts drug delivery is unknown.

Here, to address these issues, we used two-photon in vivo imaging to examine how distinct types of cerebral vessels handle drug nanocarriers in real-time, in the intact brain, in anesthetized and awake mice. We characterized the pharmacokinetics of nanoparticles targeted toward the TfR at the BBB, their intracellular trafficking patterns in vascular BECs, and transcytosis-mediated entry and transit in the brain parenchyma. We report that TfR-targeted nanoparticles bind to BECs at venules and capillaries but not at arterioles. Despite the highest association of nanoparticles to BECs in capillaries, we found that the trafficking to the brain occurs almost exclusively at post-capillary venules, with a negligible contribution of capillaries. The vascular locus of nanoparticle transport was consistent with the presence of perivascular space around post-capillary venules, which facilitates the movement of nanoparticle-sized elements in the CNS. These observations provide insight into therapeutic nanoparticle trafficking and delivery to the brain and challenge the assumed view that capillaries are the hub for effective brain transport of nanoparticle drug carriers. Thus, our findings prompt reconsideration of nanoparticle targeting strategies for improved drug delivery to the brain.

## Results

**Two-photon imaging of nanoparticles in vivo.** To study BBB-nanoparticles interactions in vivo, we used two-photon fluorescence microscopy. The brain was imaged via a cranial window over the somatosensory cortex in mice (Fig. 1a, b). The liposome nanoparticles were designed to resemble clinically approved formulations[19], i.e., consisted of a distearoylphosphatidylcholine (DSPC)/cholesterol lipid bilayer surrounding an aqueous lumen, with a polyethylene glycol (PEG) coating to ensure stability in the blood (Fig. 1c, Supplementary Table 1). Targeting of the transcytosis pathway in BECs was enabled by coupling high-affinity anti-TfR antibodies (clone RI7217) to the nanoparticle surface[20]—a targeting moiety that mediates a high level of nanoparticle binding to the brain endothelium[21,22].

Prior to imaging experiments, we validated the nanoparticle formulation and targeting strategy by encapsulating the BBB-impermeable drug cisplatin into the nanoparticles (Supplementary Table 1). The brain uptake of cisplatin was measured using inductively coupled plasma-mass spectrometry (ICP-MS)[22], and at 2 h after intravenous (i.v.) injection of the nanoparticles, cisplatin was detected in the brain only for nanoparticles functionalized with RI7217, but not for stealth (no targeting antibody), or isotype IgG antibodies (no TfR recognition) (Fig. 1d).

To study nanoparticle-BBB interactions in detail, in all subsequent experiments, we utilized two-photon imaging. The nanoparticle lipid bilayer was tagged with either Atto 550 or Atto 488 fluorophores (Fig. 1c). Both the Atto 550- and Atto 488-labeled nanoparticles (RI7-L-A550 and RI7-L-A488) were administered by single bolus injection into the bloodstream (70 nmol$_{lipid}$/g$_{animal}$), and the imaging was performed in a somatosensory cortex volume that contained all microvascular segments, i.e., arterioles, venules, and capillaries (Fig. 1e). Following the injection, the blood-circulating nanoparticles exhibited high fluorescence stability over time, indicating a low filtration rate by peripheral organs (Fig. 1f, g, Supplementary Movie 1). A small fraction of nanoparticles was sequestered by circulating leukocytes, but this did not inhibit the interaction of the leukocytes with post-capillary venules (Supplementary Fig. 1a–d, Supplementary Movie 2). Consistently, we observed no detrimental effects of nanoparticles on the brain and systemic physiology (Supplementary Fig. 1e; Supplementary Methods). Nanoparticles that circulated in the bloodstream retained their structural stability, as ascertained after laser-extravasation of nanoparticles to the brain parenchyma, i.e., temporary opening of the BBB by a short burst of the laser beam on the vessel wall (duration = 2 s, area = ~1 μm$^2$). The nanoparticles exhibited a discrete single-particle appearance without signs of agglomeration or fusion, even when extravasated after 3 h in circulation (Fig. 1h, i). Both types of nanoparticles exhibited properties of a point source signal with a Gaussian fluorescence distribution peak with average standard deviation ($\sigma$) equal to 0.290 ± 0.006 μm and 0.296 ± 0.008 μm for RI7-L-A550 and RI7-L-A488 respectively, which did not differ between distinctively labeled nanoparticles (Fig. 1j). In subsequent experiments, the nanoparticles were considered spatially separated from other sources of fluorescence when their peaks were in the distance exceeding $2\sigma$ of their fluorescence distribution profile. This distance satisfied the Rayleigh separation criterion for optical microscopy, being larger than the minimum resolved distance equaling 0.53 μm at excitation $\lambda = 870$ nm and NA$_{objective} = 1$.

Thus, the nanoparticles fulfilled all necessary requirements for in vivo imaging: efficient two-photon excitation and fluorescence

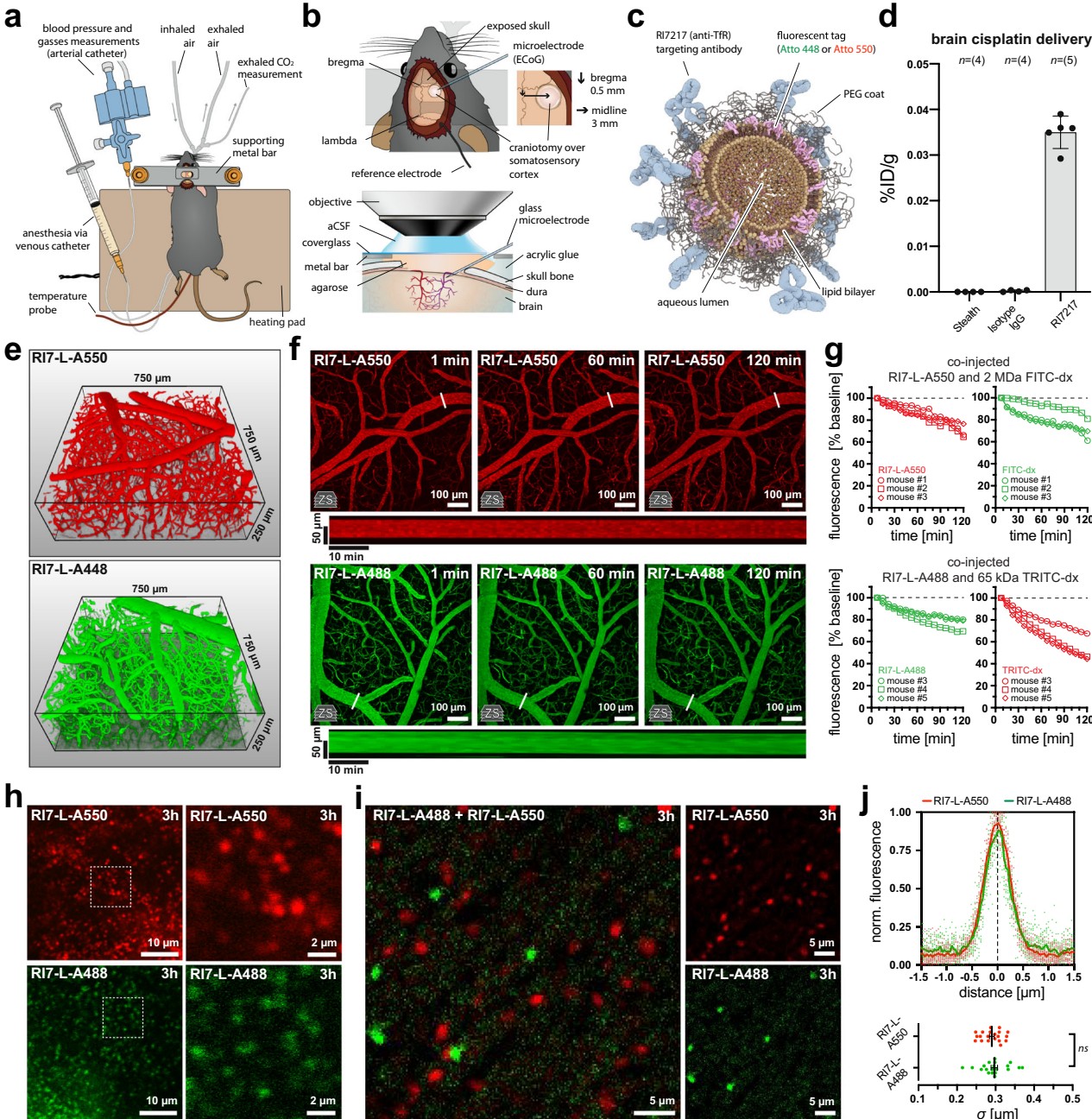

**Fig. 1 Two-photon imaging of liposome nanoparticles in vivo. a** Schematic drawing of a mouse after the preparative surgery. **b** Mouse head with the location of craniotomy and the main features of the cranial window. **c** Principal components of nanoparticles used in the study. **d** Nanoparticles functionalized with RI7217 antibody drive accumulation of cisplatin payload in the brain, in contrast to nanoparticles with isotype IgGs, and nanoparticles devoid of antibody (stealth). %ID/g is the percentage of the injected dose of nanoparticle-encapsulated cisplatin per gram brain tissue measured 2 h post i.v. injection with ICP-MS; $n_{stealth, IgG} = 4$ mice, $n_{RI7217} = 5$ mice. **e** Brain somatosensory cortex volume imaged in the study. Example 3D reconstructions show brain microvessels delineated by fluorescence of blood-circulating nanoparticles. **f** Hyperstack (Z-stack over time) images of fluorescence signal from nanoparticles after injection into the bloodstream. Time is relative to the time of injection. Long panels are kymographs of fluorescence changes measured from venules at demarked lines. 'zs' inset indicates Z-stack maximum intensity projection. See also Supplementary Movie 1. **g** Fluorescence from blood-circulating RI7-L-A550 and RI7-L-A488 exhibit a relatively high degree of stability over time, being in the range of commonly used tracers in vascular imaging studies, i.e., FITC- and TRITC-dextran. RI7-L-A550 and RI7-L-A488 signals were measured simultaneously with co-injected FITC- and TRITC-dextran, respectively. Traces represent single animals with symbols matching individual mice between corresponding imaging channels. $n = 3$ mice in each group. **h** Laser-extravasated nanoparticles into the brain parenchyma 3 h after injection into the circulation. Nanoparticles retain their discrete and homogeneous appearance even after 3 h of circulation. **i** Laser-extravasated RI7-L-A488 and RI7-LA550 into the brain parenchyma 3 h after co-injection into the circulation. Lack of merged fluorescence signal from nanoparticles indicates no liposome fusion or aggregation when in circulation or in the brain. **j** The averages of nanoparticles fluorescence profile plots after laser-extravasation into the brain. The average standard deviations ($\sigma$) of Gaussian profiles describing fluorescence signal distribution do not differ between distinctively labeled nanoparticles. $n = 20$ vs. 20 nanoparticles extravasated in 1 mouse per nanoparticle type, $p = 0.5910$, two-tailed $t$-test. All panels: ns non-significant. Data are means ± SEM.

emission, with the ability to resolve single nanoparticles, high structural stability in the circulation, and no observable detrimental effects on the brain and systemic physiology.

**Targeted nanoparticles associate to capillaries and venules, but not to arterioles.** The first step of RMT is the recruitment of a nanoparticle from the circulation to the luminal membrane of the BECs. Since in vitro BBB models fail to reproduce other vessel phenotypes than capillaries, we investigated whether TfR-targeted nanoparticle recruitment differs between distinct vascular segments in vivo. The vessels were classified as pial arterioles, penetrating arterioles, and capillaries using the anatomical features, vessel orientation, and direction of the blood flow (Supplementary Methods). In mice, the capillaries branch from ~10 μm lumen diameter vessels to reach an average ~<6 μm diameter[23,24], then converge into larger post-capillary venules with diameter >6 μm[25], that project to ascending venules or directly to pial venules[17]. RI7-L-A550 nanoparticles were co-injected into the bloodstream with 2 MDa FITC-dextran (FITC-dx) to delineate the vessel lumen. At 2 h post-injection, nanoparticles were present as numerous punctae at the vessel walls of the pial, ascending, and post-capillary venules, and capillaries (Fig. 2a–e), but did not associate to the arterial branches of the brain microvasculature (Fig. 2f). Next, we ascertained that a nanoparticle targeting moiety, i.e., antibody, is necessary for a circulating nanoparticle to be captured from the bloodstream at the luminal side of the BECs. The association was driven by the TfR-Ab binding to the TfR, and not by non-specific interactions since neither antibody-lacking stealth nanoparticles (Sth-L-A550, Supplementary Fig. 1f, g) nor isotype IgG-functionalized nanoparticles that lack TfR recognition associated to BECs (IgG-L-A550, Supplementary Fig. 1h, i). Noteworthy, the association was independent of the type of fluorescent tag, as RI7-L-A488 exhibited the same targeting properties as RI7-L-A550 (Supplementary Fig. 2a–e). In addition, mice co-injected with both nanoparticles exhibited distinct labeling at the same vessel walls from both RI7-L-A550 and RI7-L-A488 nanoparticles, attesting that the observed punctae represented single nanoparticles (Fig. 2g). Scarce presence of merged signals was attributed to overlapping fluorescence from nanoparticles separated by a distance smaller than the diffraction limit of the microscope (Fig. 2h).

**Vessel type determines nanoparticle association density and kinetics.** We then assessed whether there were differences in nanoparticle distribution and association kinetics among vessels that recruited nanoparticles from the bloodstream. BEC-associated nanoparticles were separated from blood-circulating nanoparticles by excluding the signal of circulating nanoparticles that co-localized with FITC-dx in the vessel lumen (Fig. 3a). Three-dimensional reconstructions revealed a spatially heterogeneous association of nanoparticles across the vascular interface (Fig. 3b). At 2 h post-injection, the highest density of nanoparticles was at vessel walls of capillaries, with an exponential decline along the post-capillary to pial venule axis (Fig. 3c). This was consistent with the kinetics of association being fastest in capillaries and slowest in pial venules (Fig. 3d, e, Supplementary Fig. 2f–h, Supplementary Movie 3). Of note, nanoparticle binding was ongoing even after 2 h post-injection, indicating that regardless of the vessel type, the cellular pool of available TfRs was not saturated at this time (Fig. 3e).

**Most associated nanoparticles are internalized by the endothelium.** Association of a nanoparticle to the luminal side of the vessel does not guarantee endocytosis into the BEC, and thus, the ratio of adhering to internalized nanoparticles at the BEC is informative to understand the uptake efficiency per immobilized nanoparticle. Worth highlighting, in vivo two-photon imaging provides an unbiased assessment of this ratio, compared to the conventional histological analyses, which may impact the fraction of luminally immobilized nanoparticles by direct washout, interference with receptor interaction, or by dissolving the lipid-anchored fluorophore of the nanoparticle (Supplementary Fig. 3a–c). We utilized transgenic mice expressing cytosolic green fluorescent protein (GFP) in the endothelium (Tie2-GFP) to ensure an anatomical setpoint for the analysis. This allowed for a comprehensive assessment of the BEC morphology with subcellular details in the living brain (Fig. 4a, Supplementary Fig. 3a) at different segments of the brain microvasculature (Fig. 4b). The GFP fluorescence was compatible with the imaging of the nanoparticles with no significant overlap between the fluorescence emission spectra (Fig. 4c). Noteworthy, we found that the nanoparticles associated to the BEC predominantly at higher (3+) branching orders of capillaries, exemplifying the heterogeneity of the BBB even among vessels of the same category (Fig. 4d).

At 2 h post-injection, TfR-targeted nanoparticles exhibited a non-uniform distribution in relation to the vessel wall, appearing to be internalized or to adhere to the luminal surface of the endothelium (Fig. 4e). For each analyzed vessel, we aligned the imaging plane with the vessel's long symmetry axis and extracted the fluorescence profiles of the nanoparticles and the corresponding endothelium (Fig. 4e). As a measure of separation between nanoparticles and BECs, we used the distance between fluorescence peaks ($\Delta p$) given in standard deviations of the nanoparticle fluorescence spread ($\sigma = 0.29$ μm, Fig. 1j). We categorized nanoparticles as adhering, internalized, and abluminal with the remaining nanoparticles belonging to an unresolved fraction at the luminal or abluminal side of vascular endothelium (see "Methods"), and tested our boundary conditions by visual inspection of the images (Fig. 4f). Given a wide point spread function along the Z-axis (depth), we refrained from analyzing capillaries because of their small circumference. The internalized fractions equaled 52.4%, 45.7%, and 45.4% of total nanoparticles at the BEC for main pial venules, pial venules, and ascending venules, respectively (Fig. 4f). The corresponding adhering fraction of nanoparticles was 23.6%, 30.3%, and 32.5%. Since nanoparticles located further toward the abluminal side than internalized nanoparticles had also been endocytosed by BEC at some point, the total fraction of sequestered nanoparticles was 64.2%, 54.6%, and 50.9% for main pial, pial, and ascending venules, respectively (Fig. 4f). Next, we calculated high and low estimates of internalized vs. adhering nanoparticles ratio, where high estimates counted all unresolved luminal nanoparticles as internalized, and low estimates counted all unresolved luminal nanoparticles as adhering nanoparticles (Fig. 4g). In both cases, and for all vessel types, the ratio was >1, suggesting that at 2 h post-injection, the majority of the nanoparticles recruited to the vessel walls were internalized by the BECs. Noteworthy, both high and low uptake estimates increased with the venule diameter.

**Nanoparticle movement dynamics.** Having established that the majority of the vessel-resident nanoparticles were internalized into BECs, we subsequently characterized the dynamics of intracellular nanoparticle trafficking. Once internalized, the nanoparticles exhibited a relatively high degree of motility (Fig. 5a, Supplementary Movie 4). Nanoparticles were tracked 2–3 h post-injection throughout 30 min of continuous imaging ($\Delta t$ between frames = 30 s) in vessels with the long symmetry axis aligned with the imaging plane (Fig. 5b, Supplementary Movie 5). To keep the number of nanoparticles per vessel segment consistent, we selected

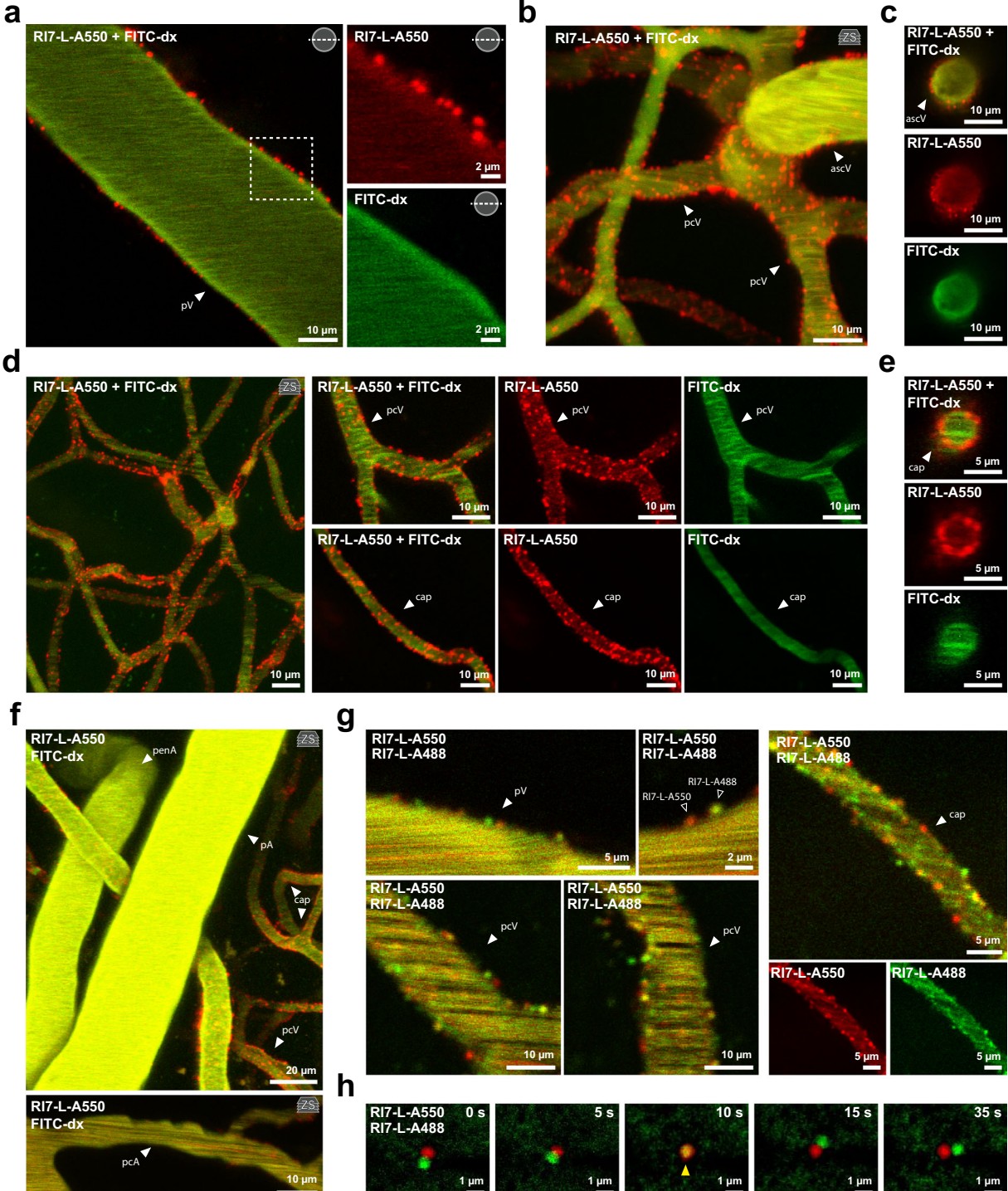

**Fig. 2 Robust association of nanoparticles to the BBB. a**–**e** Two-photon in vivo images of RI7-L-A550 nanoparticles (red) 2 h after injection into the circulation. Co-injected FITC-dx (green) delineates vessel lumen. Nanoparticles readily associate to vessel walls at pial venules, ascending venules, post-capillary venules, and capillaries. **f** In contrast to venules and capillaries, nanoparticles do not associate to arterial branches of the brain microvasculature. **g** Co-injection of both RI7-L-A550 and RI7-L-A488 reveals discrete punctae of both variants of nanoparticles at the vessel walls. **h** Time-lapse images of laser-extravasated nanoparticles in brain parenchyma. The presence of merged fluorescence signal is because of the fluorescence signal overlap when nanoparticles are in proximity to each other (arrowhead) and not due to nanoparticle fusion or exchange of fluorophores. All panels: pV pial venule, ascV ascending venule, pcV post-capillary venule, cap capillaries, penA penetrating arteriole, pA pial arteriole. 'zs' inset indicates Z-stack maximum intensity projection.

the 10 most motile nanoparticles that remained in the imaging plane from each analyzed vessel segment. The traces were aligned to the point of origin (Fig. 5c), and to avoid underestimation of movement in planar (x,y) coordinates, especially in vessels with a small circumference, we projected each trace to a vector **V** aligned

with a vessel symmetry axis and blood flow direction, both independent from the vessel curvature (Fig. 5c). To avoid underestimation of the movement along the vessel *v* axis, we omitted ascending venules because of their high-angle orientation to the imaging plane (see "Methods").

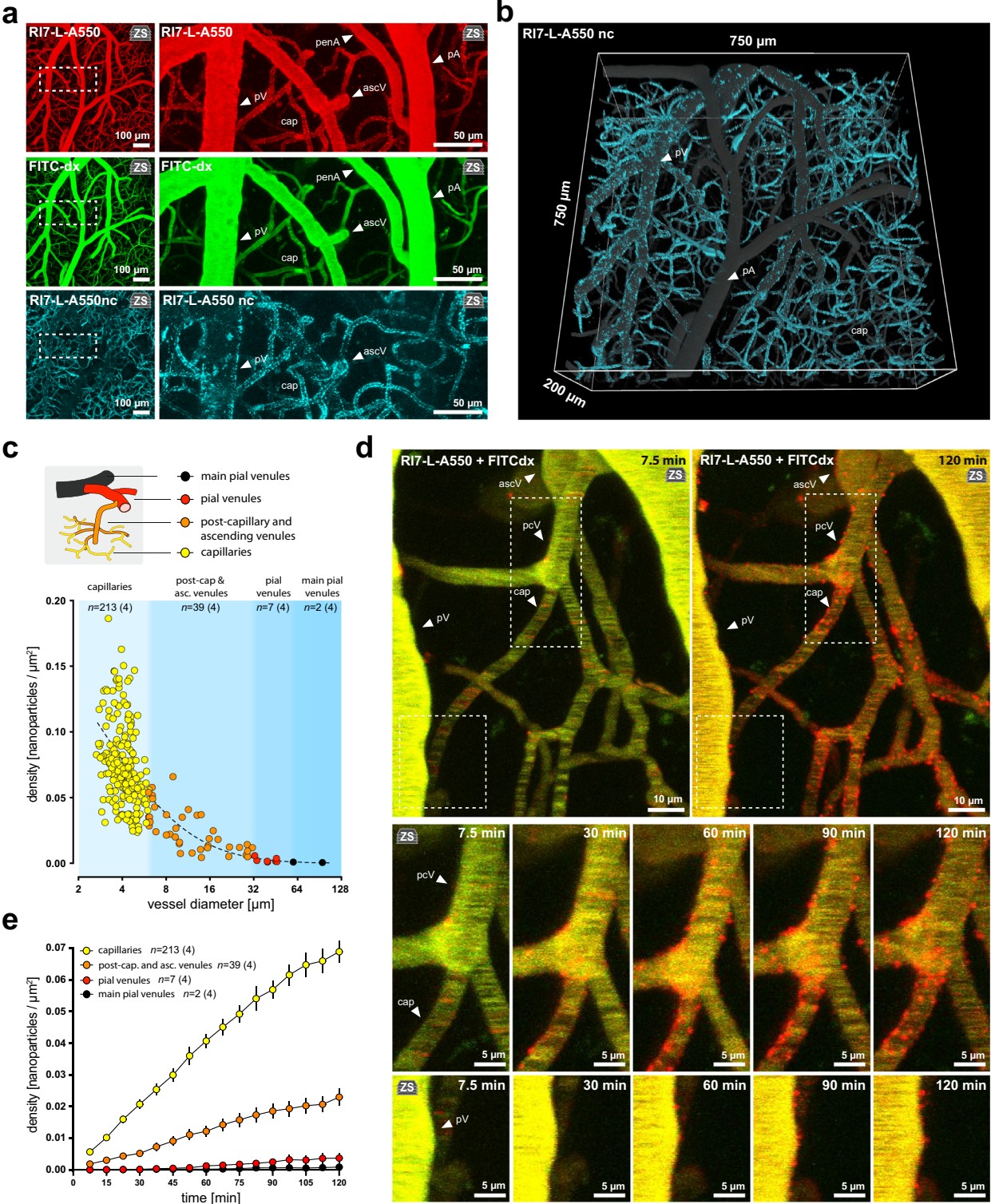

Despite anatomical differences between pial venules and capillaries (Fig. 4a, b, Supplementary Fig. 3a), the average total distance traveled by the nanoparticles did not differ between these vessel segments. However, for both pial venules and capillaries, it was ~20% shorter (~1 μm/30 min) compared to the post-capillary venules (Fig. 5d). This movement was unaffected by the direction of blood flow, as in all vessel types, the median displacement relative to the blood flow direction approximated 0 (Fig. 5e). Noteworthy, nanoparticles exhibited movement even in capillaries with stalled blood flow (Supplementary Movie 6). Thus,

neither differences in cell size and blood flow velocities at distinct vascular segments[26] nor blood flow direction influenced the intracellular movement of nanoparticles.

To assess whether the intracellular trafficking of nanoparticles occurs by random movement (e.g., diffusion), we performed a mean squared displacement (MSD) analysis of trajectories along the vessel axis $v$ (Fig. 5f). The nanoparticles exhibited significant deviation from linearity regardless of the vessel type. This indicated that the nanoparticle movement was inconsistent with diffusion in all vessel types. The directional component was

**Fig. 3 Nanoparticle association density and kinetics differ between capillaries and venules. a** The difference between RI7-L-A550 (red) and FITC-dx (green, vessel lumen) fluorescence signal shows the fraction of RI7-L-A550 nanoparticles that are non-circulating and are associated to vessel walls (RI7-L-A550nc, cyan). Non-circulating nanoparticles contour vessel boundaries in venules and capillaries, and are absent in arterioles. **b** 3D reconstruction of the signal from nanoparticles associated to endothelium 2 h post-injection in vivo. The recruited (non-circulating) nanoparticles (RI7-L-A550nc, cyan) are superimposed on the signal from nanoparticles circulating in the bloodstream (gray). The signal from RI7-L-A550nc (cyan) shows a spatially heterogeneous association of nanoparticles across the vascular interface in the living brain. **c** Nanoparticles distribution 2 h post-injection showing preferential association of nanoparticles to capillaries (yellow). Each point represents a single vessel. Nanoparticle surface density = number of nanoparticles per $\mu m^2$ vessel wall area. Blue areas demark clusters of vessels of the same type. Dashed line demarks lognormal distribution trendline. $n$ = number of vessels, where $n_{cap}$ = 213; $n_{pcV\_ascV}$ = 39; $n_{pV}$ = 7; $n_{mV}$ = 2 across 4 mice. Inset illustrates vessel hierarchy and color-coding. **d** Upper panel: hyperstack (Z-stack over time) imaging of the brain microvasculature after co-injection of RI7-L-A550 nanoparticles (red) and FITC-dx (vessel lumen, green) into circulation. Squares indicate areas magnified in lower panels. Lower panels: time-lapse images of nanoparticle association over time to vessel walls. See also Supplementary Movie 3. **e** Nanoparticles associate most rapidly to capillaries (yellow) and most slowly to pial venules (red, black). Surface density = number of nanoparticles per $\mu m^2$ vessel wall area (Supplementary Note). $n$ = number of vessels, where $n_{cap}$ = 213; $n_{pcV\_ascV}$ = 39; $n_{pV}$ = 7; $n_{mV}$ = 2 across 4 mice. All panels: pV pial venule, ascV ascending venule, pcV post-capillary venule, cap capillaries, penA penetrating arteriole, pA pial arteriole. 'zs' inset indicates Z-stack maximum intensity projection. Data are means ± SEM.

present in capillaries and post-capillary venules, where $MSDv(\Delta t)$ exceeded values predicted by normal diffusion (linear fit), but was not apparent in pial vessels that exhibited an anomalous average trace. Although the MSD analysis does not disclose the underlying biological background, it suggested that the internalized nanoparticles do not move randomly but rather via coordinated intracellular trafficking both in capillaries and post-capillary venules.

**Nanoparticles distribute to endothelial perinuclear areas in venules but not in capillaries.** Next, we analyzed the subcellular distribution of internalized nanoparticles. The nanoparticles localized over time to the perinuclear region of BECs in venules (Fig. 5g), but this was not observed in capillaries (Fig. 5h). We quantified the spatial distribution of nanoparticles 3 h post-injection in venules by measuring their Euclidean distances from the geometric center of the nucleus (Fig. 5i). We refrained from measurements in capillaries because of systematic underestimation of nanoparticle distances from the nucleus due to high vessel curvature. Our data shows that the highest probability for finding a nanoparticle in venules was at distances of 0.5–2.5 μm from the nucleus boundary, and with numbers decreasing at intermediate (2.5–5 μm) and distal (>5 μm) regions (Fig. 5j, k). We observed no clustering of nanoparticles at the endothelium perimeter (Fig. 5l), indicating that nanoparticles do not wedge between adjacent endothelial cells, or stall in the cytosol areas with dense cytoskeleton elements that support cell contact sites.

**Lack of perivascular space impedes nanoparticle brain transit.** The possibility of nanoparticle transcytosis across the BBB is still disputed, especially for high-affinity binding to the TfR[27–31]. Here, in contrast to capillaries, nanoparticles in venules exhibited clearly observable translocation toward the brain with distances exceeding the endothelial thickness (Fig. 6a). Our imaging revealed the occurrence and dynamics of transcytosis, where nanoparticles, which associated to the venule walls, slowed down, but once transcytosed, they rapidly progressed in the perivascular space (Fig. 6b, Supplementary Movie 7) and further into the brain (Fig. 6c, Supplementary Movie 8).

Given that capillaries are devoid of perivascular space[17,25], and the transcytosis-mediated entry to the brain occurred in venules, we tested whether the lack of perivascular space impedes nanoparticle movement at the abluminal side of capillaries in comparison to post-capillary venules. We laser-extravasated nanoparticles from the bloodstream at capillaries and venules. Following extravasation, the nanoparticles in proximity to venules rapidly migrated from the imaging plane,

consistent with the movement observed after transcytosis. In contrast, the nanoparticles extravasated at capillaries exhibited severe movement impairment and appeared stationary (Fig. 6d, Supplementary Movie 9). Both the total distance traveled and displacement of nanoparticles were significantly higher in the brain tissue surrounding venules than capillaries (Fig. 6e). This suggests that the lack of perivascular space impedes the movement of nanoparticles on the abluminal side of the capillaries.

Of note, we also detected rare events where blood-borne cells, which had sequestered the circulating nanoparticles, crossed the BECs and entered the perivascular spaces of the venules (Supplementary Movie S10), consistent with the location of immune-cell trafficking in the brain[17].

**Post-capillary venules are the key locus of transcytosis-mediated brain entry.** Immunohistochemical analysis enables the assessment of distinct brain regions, but chemical processing of the tissue collapses perivascular spaces[32,33], and obliterates the discrete signal of nanoparticles in the brain (Supplementary Fig. 3c), providing no accurate information on the nanoparticle location in the perivascular space (Supplementary Fig. 3d).

Given these methodological challenges, along with the rarity of transcytosis events and a limited span of acute imaging experiments (~4 h), we next performed long-term two-photon imaging on awake and movement-unrestricted mice with chronic cranial window implants (Fig. 6f). We imaged the somatosensory cortex in the brains of awake Tie2-GFP mice 10-day post-surgery, then reassessed the same loci 2 days later, and injected RI7-L-A550 into the bloodstream (Fig. 6g). Subsequently, we imaged the animals again 30 min post-injection to ensure that the endothelium remained structurally intact and examined the same region at 1 and 2 days post-injection (Fig. 6g). At 1 day post-injection, the animals exhibited transcytosed nanoparticles at the abluminal side of the endothelium at venules, with no significant nanoparticle presence in the brain in proximity to capillaries (Fig. 6g). The amount of nanoparticles in the brain decreased between 1 and 2 days post-injection, but was still apparent, indicating high retention (Fig. 6g, h). In contrast to RI7-L-A550, the stealth (non-targeted) nanoparticles were absent in the brain parenchyma, indicating no passive entry of nanoparticles across the BBB to the brain (Supplementary Fig. 4).

To quantify the distribution of nanoparticles, we measured the distances of individual nanoparticles from the nearest vessel, considering the vessel type and diameter. To avoid underestimating a nanoparticle distance from a vessel wall, we only counted the nanoparticles located at the imaging plane aligned to

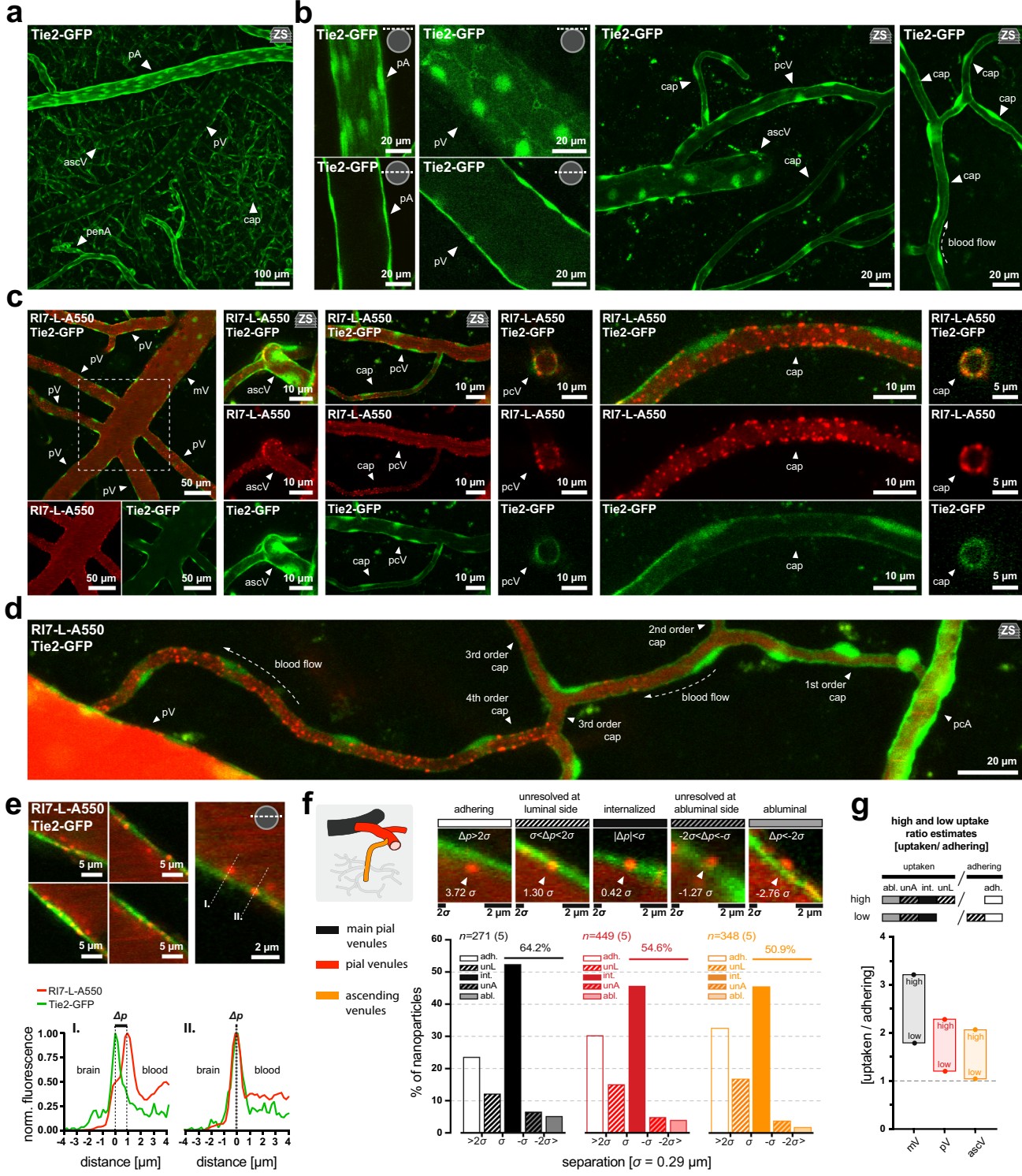

the vessel's long symmetry axis (Fig. 6j). Although the amount of nanoparticles that co-localized with endothelium was comparable between capillaries and venules (3.6% vs. 5.4% of the total count, respectively), the entry of nanoparticles occurred mainly at post-capillary and ascending venules (76.5% of the total count), with only a small fraction of nanoparticles present in the brain in proximity to capillaries (1.5% of a total count). Pial venules belong to leptomeningeal vessels and, by definition, are not considered a part of the BBB. Nonetheless, we also observed nanoparticles in proximity to pial venules (13.0% of the total count) but not in their endothelial cells (0% of a total count).

Overall, these results indicate that transcytosis-mediated delivery of nanoparticles to the brain occurs almost exclusively at venules, with a negligible contribution of capillaries (Fig. 7).

## Discussion

For nearly two decades, drug delivery studies have faced methodological challenges to determine how nanoparticles penetrate the BBB and whether there are differences in how distinct brain vessel types handle nanoparticle drug carriers. Here, we combined two-photon microscopy in vivo with drug carrier nanotechnology

**Fig. 4 Most associated nanoparticles are taken up by the endothelium. a** Z-stack projection of the brain cortical microvasculature endothelium imaged in Tie2-GFP mice in vivo. **b** Distinct morphology of brain endothelial cells at different vascular segments in vivo. Note the clear presence of cell contact sites and bright nuclei. **c** Examples of simultaneous imaging of brain endothelium (green) with RI7-L-A550 nanoparticles (red) 2 h after injection into the circulation. **d** Nanoparticles associate predominantly to high branching order capillaries. **e** Square panels: A fraction of nanoparticles is restricted from entering the brain endothelium and reside on the luminal side. Right panel: example of nanoparticles (red) at two distinct locations (I., II.) in relation to the endothelium (Tie2-GFP, green). Dashed lines indicate the axes of fluorescence profiles in the lower panel. Lower panel: fluorescence profiles of nanoparticles with low (I.) and high (II.) signal overlap with endothelium. $\Delta p$ demarks nanoparticle and endothelium signal peaks separation. **f** Upper row: Examples of nanoparticle classification based on $\Delta p$. Numbers at arrowheads represent $\Delta p$ in $\sigma$ units ($\sigma = 0.29 \, \mu m$). Lower graph: percentage distribution of adhering (adh.), internalized (int.), unresolved at luminal (unL) or abluminal side (unA) nanoparticles, and nanoparticles found at the abluminal side of the endothelium (abl.). Values over horizontal lines are the sum of underneath bins. $n$ = number of nanoparticles, where $n_{mV}$ = 217; $n_{pV}$ = 449; $n_{ascV}$ = 348 across 5 mice. Inset illustrates vessel hierarchy and color-coding. **g** Nanoparticle uptake estimates with high and low estimate counting all unresolved nanoparticles on luminal side (unL) as internalized (int.), or as adhering (adh.) nanoparticles, respectively. All panels: Image insets denote the position of the imaging plane relative to a vessel perimeter. 'zs' inset indicates Z-stack maximum intensity projection. pV pial venule, ascV ascending venule, pcV post-capillary venule, cap capillaries, pcA pre-capillary arteriole, penA penetrating arteriole, pA pial arteriole.

to provide direct insight into liposome nanoparticle delivery through all stages of transport across the BBB to the brain. We reveal that the post-capillary venules are primary mediators of nanoparticle delivery to the brain parenchyma with negligible contribution from capillaries.

Although anti-TfR antibodies can cross the BBB[13,34,35], there is a fundamental disagreement as to whether the TfR-targeted nanoparticles can also enter the brain. The claim that TfR can mediate transcytosis of whole nanoparticles[27,28,31] is met by a conflicting view, where delivery occurs via a release of cargo due to the fusion of lipid-based nanoparticles with the endothelial membrane[29]. Here, we provide direct evidence of TfR-mediated transcytosis of nanoparticles across the endothelial cells and their subsequent transit in the brain parenchyma.

The current mechanistic understanding of endothelial transport comes primarily from in vitro BBB culture models, which do not preserve vascular segment heterogeneity and do not reflect well the molecular landscape of BECs in vivo. Only recently, the aspect of brain microvascular zonation was explored at the level of receptor protein expression[36], transcriptomics[37–39], and transport modulation[40]. We show that the binding of nanoparticles to the endothelium was non-uniform and rapidly declined along the capillary to venule axis, consistent with TfR distribution at the level of the receptor protein from early studies[41,42] and recent complementary data to single-cell transcriptomics[37]. TfR is a recycling receptor with ~10% of the total receptor pool present at the surface of BECs[43]. This level of surface available receptors was sufficient to maintain steady and ongoing recruitment of TfR-targeted nanoparticles to the vessel wall without saturation even at the high densities in capillaries. Although recent gene expression analyses revealed the presence of TfR mRNA (TFRC transcript) in arterioles[37,39], we detected no nanoparticles at this vascular segment. One intriguing possibility is that if the TFRC transcript is present in arterioles and is translated to the receptor protein, this pool of TfR may be functionally excluded from the uptake of nanoparticles from the bloodstream. An important consideration is what other BEC features along the vascular segments might affect the nanoparticle binding to TfR. Given that brain vasculature hemodynamics differs between arterioles, capillaries, and venules[26,44], it is plausible that the slowest blood velocity in capillaries may increase the time of interaction of nanoparticles with TfR, thus increasing the probability of nanoparticle binding. However, we observed no preferential association of nanoparticles at branching points or at vessels with high tortuosity, and no increase in binding of nanoparticles when blood-borne cells were temporarily obstructing the nanoparticles flow. Thus, although blood pressure, flow, and velocity might influence binding, they may not play a decisive role in nanoparticle distribution, which follows

more closely the vascular gradient of *TFRC* expression[37–39]. Conversely, the colloidal glycocalyx at the BEC luminal side may repel negatively charged nanoconstructs or potentially bury macromolecules within its matrix, hereby impairing endocytosis[45,46]. Indeed, we detected that the internalization of nanoparticles was not an all-or-nothing phenomenon, with at most half of the total nanoparticles recruited from the blood circulation being restricted to the luminal side of BEC. Although enzymatic shedding of glycocalyx does not improve BEC uptake of positively charged nanoparticles[45], we cannot exclude that glycocalyx might affect binding or uptake of negatively charged particles, such as the nanoparticles used herein.

A key question is why the highest association of nanoparticles in capillaries did not translate to the vascular location of the highest nanoparticle brain entry. This question can be approached from two sides; one where nanoparticle transcytosis in capillaries is negligible compared to post-capillary venules, and one where following transcytosis, the entry of nanoparticles to the brain at capillaries is impaired due to the absence of a perivascular space.

Regarding transcytosis, our data suggest that its incidence had to be low compared to post-capillary venules. Despite the highest binding of nanoparticles to capillaries and high retention of nanoparticles in the brain parenchyma, there was no significant nanoparticle presence at capillary walls at 24 and 48 h post-injection compared to post-capillary venules. Had the transcytosis occurred to a similar degree in all vessel segments, a larger nanoparticle fraction would be associated with the capillaries. Else, the transcytosed nanoparticles at capillaries would have to be re-uptaken into the BEC and further processed by lysosomes or released into the blood, similar to previously proposed mechanisms[16]. It is unlikely that nanoparticles were transcytosed at capillaries and were subsequently relocated along the capillary wall into the perivascular space of venules by the interstitial fluid flow because the hydraulic resistance around the capillaries is too high for interstitial fluid flow[47]. The negligible presence of nanoparticles at capillaries also cannot be explained solely by impaired dissociation of antibody-functionalized nanoparticles from TfR during exocytosis[11,35], as nanoparticles would also be nearly absent at venules. Instead, the difference in trafficking between capillaries and venules observed herein may be closely associated with normal vessel function. It is well established that transendothelial transport of endogenous protein receptor ligands (or antibody-based medicines) occurs at the capillary level[1]. Conversely, post-capillary venules mediate transendothelial transport of immune cells surveilling the brain[17]. It could be speculated that venular endothelial cells may contain intracellular machinery that can better handle large entities like the nanoparticles used in the study.

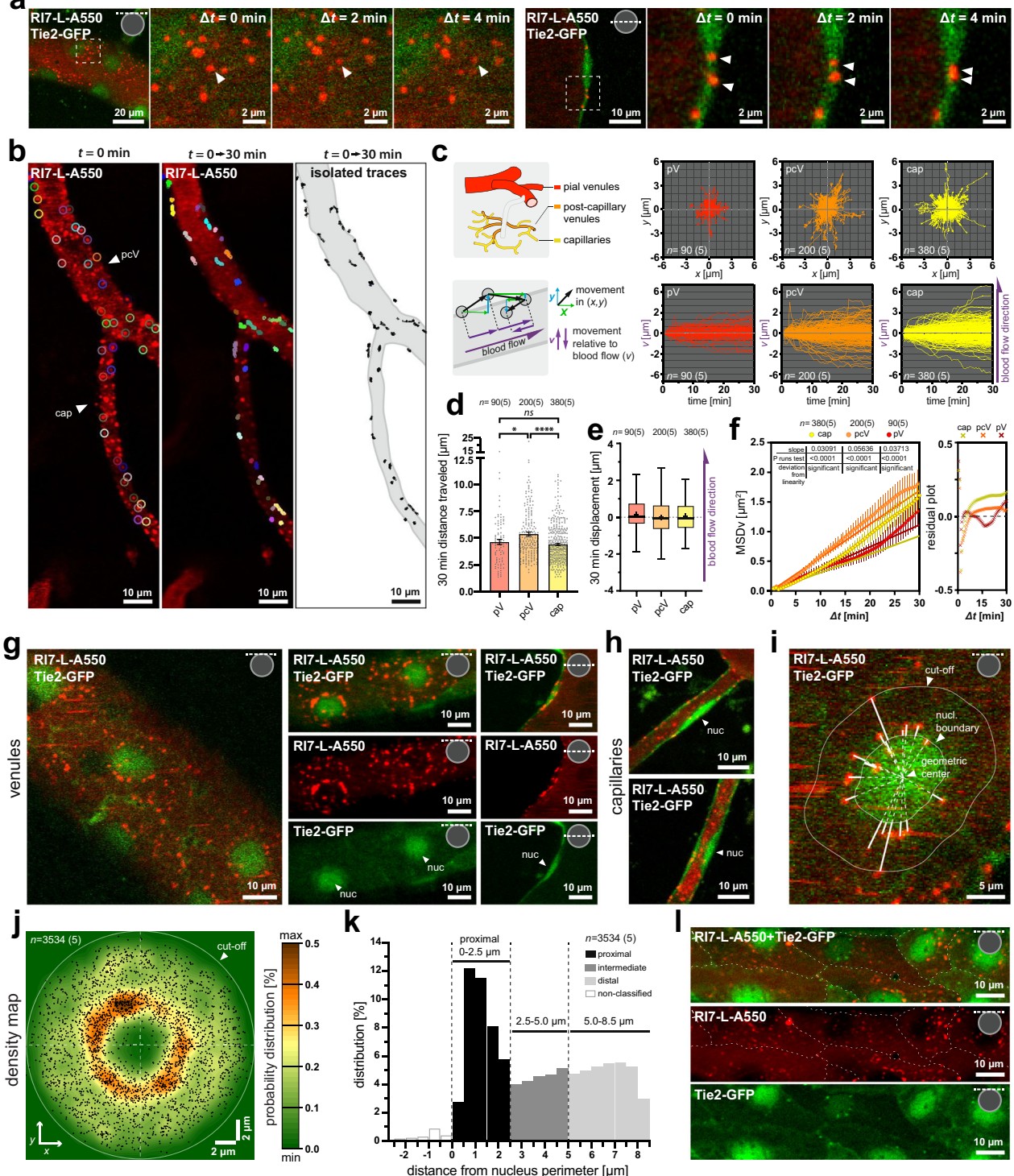

Regarding perivascular spaces, the microanatomy of the tissue surrounding microvessels may be the next factor influencing the successful nanoparticle delivery into the brain. Large and post-capillary venules are surrounded by perivascular spaces located between the endothelial basement membrane and astrocyte glia limitans. In contrast, capillaries are devoid of perivascular space and are in direct contact with the brain parenchyma because both basement membranes are fused[17,48]. This principal anatomical feature is present in both murine and human brains[49]. Our laser-extravasation data demonstrated that the nanoparticles in the perivascular space move relatively freely, as opposed to the brain

parenchyma. This is in agreement with the perivascular space being the route of least resistance, as recently observed for nanoparticles after the infusion into the brain[32], which may explain why perivascular space facilitates the distribution of therapeutics[50]. It is plausible that nanoparticles are more likely to transcytose and progress to the brain via the route of the least resistance, i.e., into the perivascular space of venules, instead of entering the more restrictive compartment, i.e., the brain parenchyma at the capillary segment[18,51]. The presence of a perivascular space may facilitate the egress of nanoparticles from the basement membrane in post-capillary venules, and its lack may

**Fig. 5 Vascular differences in nanoparticle motility and subcellular distribution. a** Time-lapse imaging of the vessel wall surface (left panel) and across the vessel (right panel). Arrowheads indicate moving nanoparticles. See also Supplementary Movie 4. **b** Nanoparticle tracking. Left panel: circles outline selected nanoparticles. Middle panel: nanoparticle movement during 30 min of continuous imaging. Right panel: isolated movement traces (black) with contours delineating microvessels (gray). See also Supplementary Movie 5. **c** Upper left inset: vessel hierarchy and color-coding. Lower left inset: translation of nanoparticle movement from $(x,y)$ to a $(v)$ coordinate aligned with the vessel long symmetry axis and direction of the blood flow. Right panels: Nanoparticle traces in $(x,y)$ and corresponding $(v)$ coordinate. $v > 0$ indicates the movement along, and $v < 0$ against the blood flow direction. **d** Nanoparticles traverse longer distances in post-capillary venules (orange), compared to pial venules and capillaries (red, yellow; *$p = 0.0302$, ****$p < 0.0001$, respectively; two-tailed $t$-tests with Bonferroni post hoc correction). Time span = 30 min. Data are means ± SEM. **e** Blood flow direction does not affect the nanoparticle displacement. Boxplots show medians with IQR, whiskers extend between 5th and 95th percentile range. **f** Deviation of MSDv($\Delta t$) from linearity indicates movement inconsistent with diffusion predicted by the linear fit (all groups ****$p < 0.0001$, Wald–Wolfowitz runs test). Data are means ± SEM. **c**–**f** $n$ = number of nanoparticles, where $n_{pV} = 90$; $n_{pcV} = 200$; $n_{cap} = 380$ across 5 mice. **g** In venules, nanoparticles distribute preferentially to perinuclear areas. Images collected 3 h post-injection. **h** Capillaries do not exhibit a preferential perinuclear distribution of nanoparticles. **i** Measurement of nanoparticles location in relation to nucleus perimeter 3 h post-injection. **j** Kernel density map of nanoparticle distribution in relation to the geometric center of the nucleus. Kernel = $2\sigma$ (=0.58 µm). The heat-map represents the probability of nanoparticle presence at a given coordinate. **k** Percentage distribution of nanoparticles in relation to nucleus perimeter. Non-classified are nanoparticles overlapping with the nucleus. **j**, **k** $n$ = 3534 nanoparticles across 5 mice. **l** Nanoparticles do not distribute to the endothelial cells boundaries or contact sites (dashed lines). All panels: Image insets denote the imaging plane in relation to a vessel perimeter. pV pial venule, pcV post-capillary venule, cap capillary, nuc nucleus, MSDv($\Delta t$) mean squared displacement in $(v)$ coordinate.

---

create a higher resistance environment impeding brain entry. This is consistent with previous ex vivo findings of nanoparticles being restricted from progressing further into the brain at capillary segments of the brain vascular network[16,52].

As such, while the explanation for the difference in nanoparticle brain entry is likely multifaceted, we suggest that these two mechanisms are not mutually exclusive and that both intracellular sorting and availability of a perivascular space may independently influence the nanoparticle delivery to the brain.

Notably, in contrast to capillaries, post-capillary and pial venules exhibited the preferential distribution of nanoparticles to perinuclear areas. This corresponded to the location of both late endosomes, and lysosomes downstream in the trafficking route[53,54], which suggests that most nanoparticles converge with the lysosomal degradation pathway[55], similarly to high-affinity antibodies and nanoparticles targeting TfR[56,57]. However, a fraction of perinuclear nanoparticles might also undergo endosomal recycling, typically located in proximity to the microtubule organizing center, at the spot close to the nucleus membrane[53,58]. This supports the notion that despite segmental differences in RMT dynamics, the microanatomy of the brain surrounding microvessels may play a decisive role in the successful brain entry of nanoparticles.

Liposome nanoparticle formulations tested for drug delivery range between 50 and 275 nm[59]. The size is known to affect, e.g., the nanoparticle uptake in a tissue-dependent manner[60], and it is likely that it may affect intracellular sorting during transcytosis and subsequent migration and distribution in the brain parenchyma. However, the sheer size may not be the sole determinant of the efficacy of nanoparticle delivery, as other parameters that go along with the changes in size may be of importance, e.g., nanoparticle rigidity which is related to its lipid composition (i.e., saturated vs. non-saturated lipids ratio)[61]. The major concern of nanoparticle-based drug delivery systems is the ability of transcytosed nanoparticles to progress further in the brain parenchyma within the extracellular space (ECS). Most transcytosed nanoparticles in our study were observed in perivascular areas, even after 48 h post-injection, which indicates high perivascular retention. However, we also observed nanoparticles at distances from post-capillary venules corresponding to the location of neuropil, indicating successful passage into the ECS. In vivo diffusion experiments estimate the ECS to be in the range of 38–64 nm[62], although recent super-resolution imaging revealed ECS clefts of ~100 nm[63]. These dimensions may not restrict the movement of large proteins, such as antibodies, but may preclude

the significantly larger nanoparticles from entering and progressing within the ECS[62,64]. How ~130 nm nanoparticles traveled within the ECS is unclear, but studies performed in human and rat brain tissue ex vivo as well as in living brains of mice in vivo show the process to be highly dependent on the level of PEGylation[64].

Improvements of the transport efficiency might be obtained by re-designing the nanoparticles to comply with the notion that low/intermediate affinity or avidity binding to receptors at the BBB is superior to the high-affinity antibody-based transport used in this study. While this concept was primarily investigated for antibody transport across the BBB, new evidence suggests that it may also impact nanoparticle transport based on targeting of the TfR[21,22], or other well-known brain drug delivery targets[65]. This has uncovered an important role of sorting tubules and the protein regulator of membrane curvature, syndapin-2, in BEC transcytosis[65], which will likely define a novel path for future brain drug delivery strategies. Nanoparticles are presumably less effective than antibodies concerning the brain entry but can encapsulate a significantly larger amount of drugs with various biochemical properties[9–11]. Therefore, they represent an important avenue for therapeutics, with a large number of preclinical evaluations and ongoing clinical trials[8,12,13], and liposomes mono-targeted to TfR outperform other RMT delivery targets in the brain[30]. Nonetheless, our findings may also be of special relevance for other targets explored for RMT-mediated nanoparticle drug delivery. For instance, the folate receptor (Folr1), despite low expression levels, may be better suited for nanoparticle delivery, being present predominantly in venules, than, e.g., lactoferrin receptor (Ltf) being expressed almost exclusively in capillaries[37]. Furthermore, while TfR is most abundant in capillaries, the recent single-cell transcriptomic analysis revealed a gradient of various BBB transporters along the vascular tree, with sparse presence in arterioles, and the highest fraction in venules, supporting the notion that venules may be in general more predisposed to mediate transcytosis than other vascular segments[36].

In contrast to parenchymal microvessels, the vasculature in the dura does not form the BBB[17,66]. Contrary to a recent study[67], we detected no TfR-targeted nanoparticles in the dura. However, we did observe blood-borne immune cells that infiltrated the brain while carrying nanoparticles that were hijacked from the bloodstream, and this immune cell infiltration occurred exclusively in venules. Thus, a fraction of nanoparticles could also enter the CNS in a manner independent of the BBB RMT. The leakage of

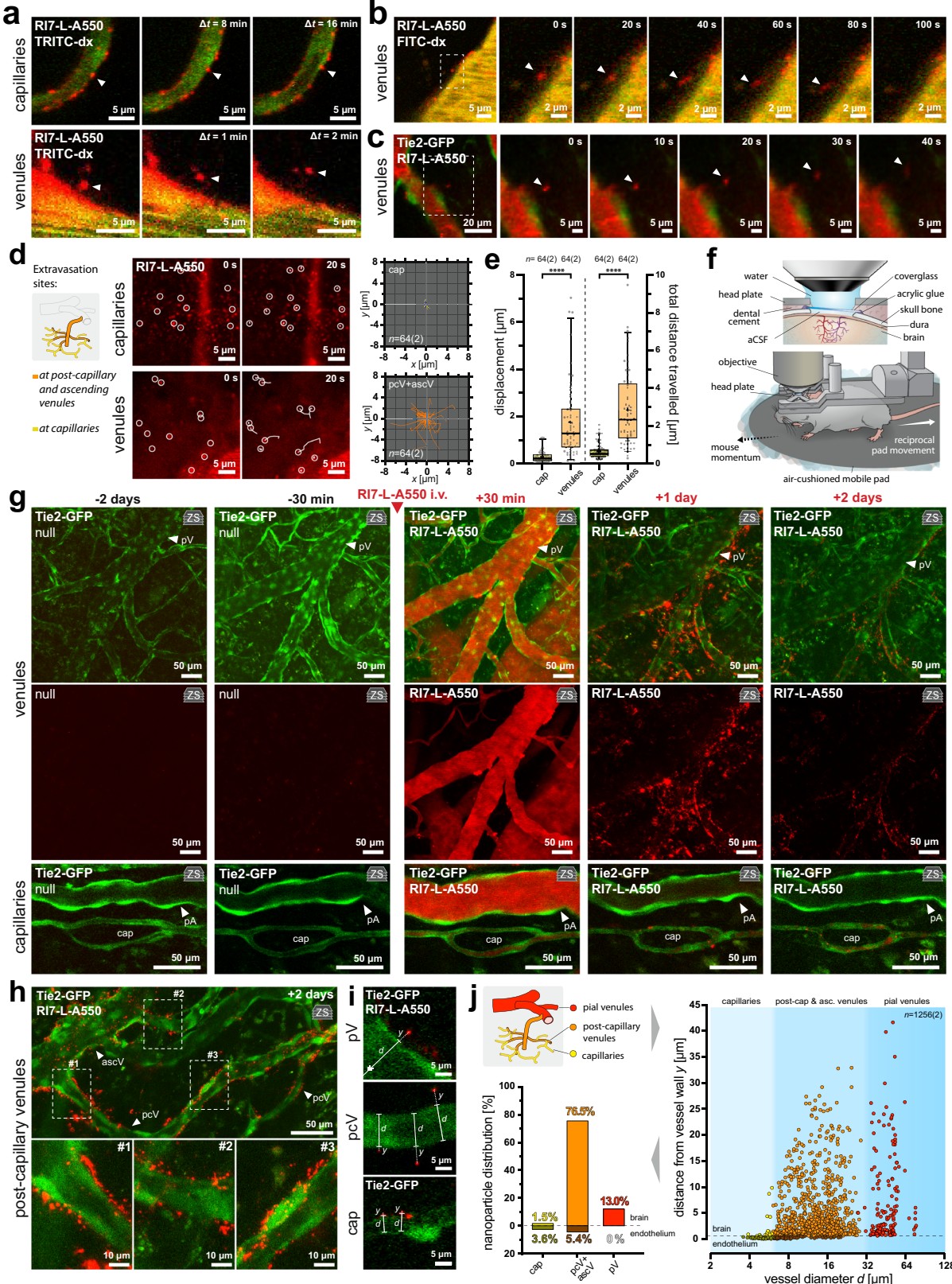

large blood-borne macromolecules can reportedly occur via wedging of leukocytes between endothelial cells in venules but not in capillaries[68]. However, this phenomenon is associated with autoimmune reactions[68], and our data showed no clustering of nanoparticles at BEC contact sites. Although TfR-targeted nanoparticles entered the brain via the post-capillary vascular segment, stalling of substantial quantities of nanoparticles in capillaries may overburden and distort the intracellular trafficking mechanisms in capillary endothelium[13]. This may be of particular relevance regarding the safety of nanoparticle-based therapies. It is plausible that nanoparticles stalled in capillaries, when loaded with pharmacologically active cargo, may eventually release high

**Fig. 6 Post-capillary venules are the key locus for transcytosis-mediated nanoparticle entry to the brain. a** Nanoparticle movement is restricted to the vessel boundary in capillaries but not in venules. **b** Real-time imaging of nanoparticle transcytosis to the brain. See also Supplementary Movie 7. **c** Nanoparticle progression in the brain following transcytosis. See also Supplementary Movie 8. **d** Nanoparticle movement traces in the brain following laser extravasation at capillaries (yellow) and venules (orange). Inset illustrates vessel hierarchy and color-coding. See also Supplementary Movie 9. **e** Nanoparticles exhibit a relatively high degree of perivascular movement at venules (orange) compared to capillaries (yellow), where the nanoparticle progression is restricted ($n = 64$ vs. 64 nanoparticles across 2 mice, ****$p < 0.0001$, two-tailed Mann–Whitney tests with Bonferroni post hoc correction). Boxplots show medians with IQR, whiskers extend between 5–95th percentile range. **f** Upper drawing: Craniotomy for chronic two-photon imaging. Lower drawing: Microscope stage with a movement-unrestricted awake animal. The objective is stationary, and the air-pressurized pad reacts reciprocally to the mouse movement. **g** Long-term imaging of nanoparticle delivery to the brain parenchyma. Upper rows: Nanoparticles are transcytosed to the brain at venular segments; Lower row: No apparent nanoparticle presence in the brain at capillaries and arterioles. **h** High-resolution images of nanoparticles after transcytosis in post-capillary venules 2 days post-injection. **i** Measurement of nanoparticles distances ($y$) to the closest vessel with a given diameter ($d$). **j** Nanoparticle distribution in the brain 2 days post-injection. Inset illustrates vessel hierarchy and color-coding. Right panel: nanoparticles distribution with respect to the vessel type, distance from the nearest vessel, and the vessel diameter. Each point represents a single nanoparticle, $n = 1256$ nanoparticles across 2 mice. Left panel: corresponding percentage distribution of nanoparticles. Dashed lines separate nanoparticles found in the brain at capillaries (yellow), post-capillary and ascending venules (orange), and pial venules (red); from nanoparticles in the endothelium at capillaries (dark yellow), and post-capillary and ascending venules (brown). All panels: 'zs' inset indicates Z-stack maximum intensity projection. pA pial artery, pV pial venule, ascV ascending venule, pcV post-capillary venule, cap capillary.

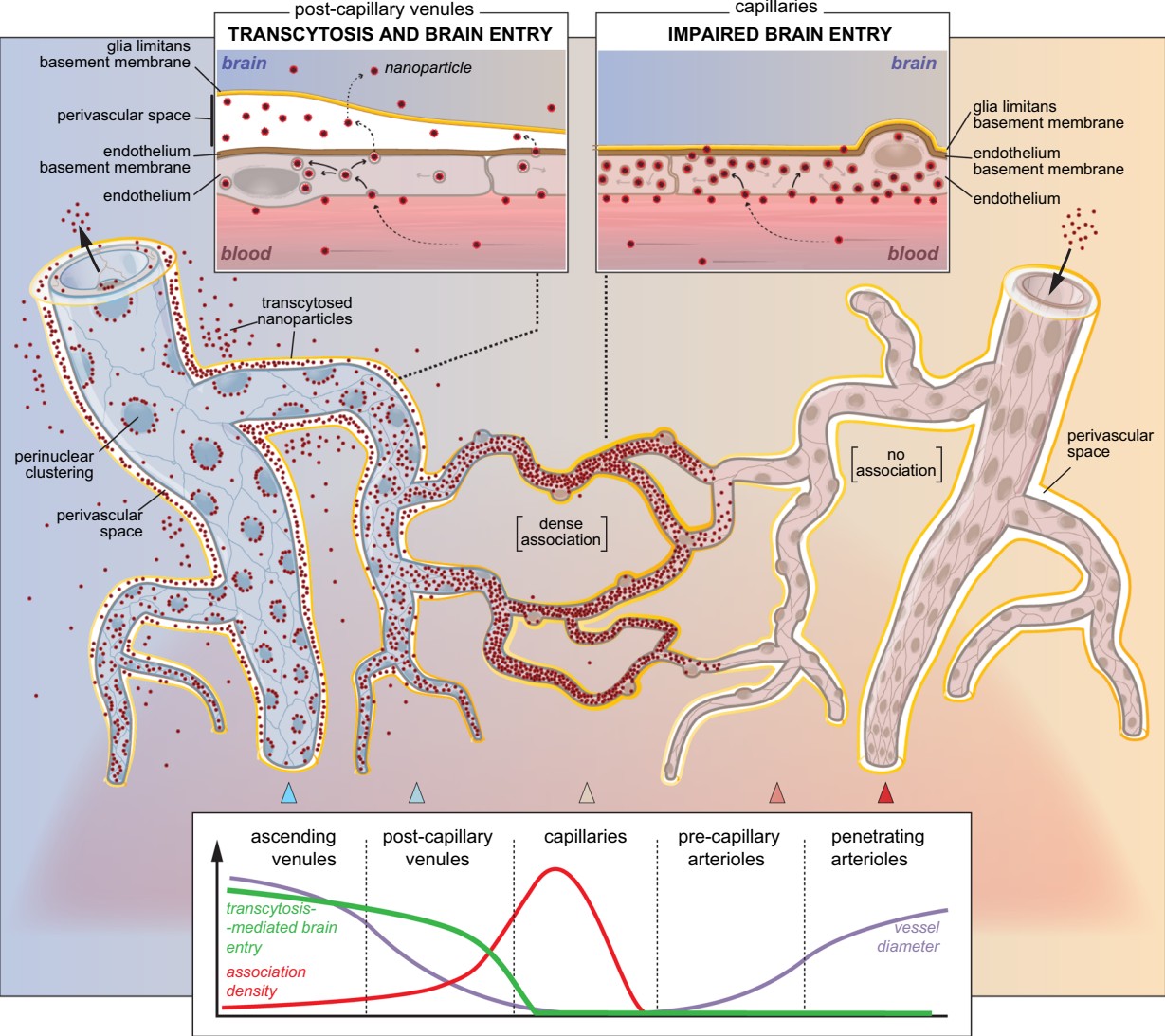

**Fig. 7 Working model of transcytosis-mediated nanoparticle delivery to the brain.** Despite the highest association to capillaries, the TfR-mediated delivery of liposome nanoparticles occurs almost exclusively in post-capillary venules, and is negligible in capillaries, where the lack of perivascular space impedes the progression of nanoliposomes into the brain parenchyma.

concentrations of the payload within capillary endothelium, exerting negative side effects on vascular function.

Last, our methodological platform allows to describe the nanoparticles in the living brain at microscale resolution in live, and in awake animals. It is suitable for examining the potential effects of changes in transcytosis on large molecule therapeutics delivery in the aged brain, recently reported to exhibit a substantial decline in RMT[36]; and in disease states, e.g., during ischemia-induced imbalance in transcytosis[69], reduction of perivascular spaces[32], or ECS changes in edema[63]. Our findings may also help to avoid pitfalls in the design of drug delivery systems, e.g., in Alzheimer's disease, where Aβ may deposit along periarteriolar, rather than venular perivascular space[70].

In summary, we identified post-capillary venules as the key site for transcytosis-mediated brain entry of nanoparticles. Our results provide a vascular zonation map of TfR-targeted nanoparticle trafficking that may aid efforts to develop efficient and safer therapeutic approaches. In contrast to capillaries aimed routinely for drug design strategies, post-capillary venules as an accessible part of the vascular segment are more apt for nanoparticle delivery across the BBB.

## Methods

**Animals**. All animal experiments were approved by The Danish National Committee on Health Research Ethics following the guidelines established by the European Council's Convention for the Protection of Vertebrate Animals Used for Experimental and Other Scientific Purposes (reference numbers: #2019-15-0201-01655; #2019-15-0201-01658; #2016-15-0201-00920), and were in compliance with ARRIVE guidelines. We used wild-type C57Bl/6 mice, age 5–7 months (23–31 g) and age-matched (25–32 g) homozygous Tg(TIE2GFP)287Sato/J transgenic reporter mice (Tie2-GFP mice, #003658, The Jackson Laboratory) expressing GFP under the endothelial-specific receptor tyrosine kinase (Tie2) promoter[71]. All animals were housed in ventilated cages under a 12 h light/12 h dark cycle, at 50 ± 10% relative humidity, at room temperature, with ad libitum access to food and water. The animal housing facility has been accredited by the Association for Assessment and Accreditation of Laboratory Animal Care (AAALAC), and the Federation of Laboratory Animal Science Associations (FELASA).

**Animal preparation for acute imaging**. Surgery was performed as previously described, with minor modifications[72]. Briefly, animals were anesthetized by intraperitoneal (i.p.) injection of xylazine (10 µg/g$_{animal}$) and ketamine (60 µg/g, then 30 µg/g$_{animal}$, at 20–25 min intervals). A tracheotomy was performed for mechanical respiration (180–220 µL volume at 190–240 strokes/min; MiniVent Type 845, Harvard Apparatus) with O$_2$-supplemented air (1.5–2 mL/min). Two catheters were inserted, one into the left femoral artery for injection of compounds and nanoparticles, and for monitoring mean arterial blood pressure (MABP; Pressure Monitor BP-1, World Precision Instruments), and the other into the femoral vein for anesthesia infusion during imaging. The animal was turned to the prone position and the scalp was removed. The periosteum was removed with a FeCl$_3$-soaked cotton bud, and the exposed skull was glued (Loctite Adhesives) to a custom-made metal head plate. A craniotomy was performed over the right somatosensory cortex (3 mm lateral, 0.5 mm posterior to bregma; Ø = 4 mm; 4500 rpm dental drill). The bone flap was carefully lifted, the dura removed, and 1% agarose (type III-A, Sigma-Aldrich) in artificial cerebrospinal fluid (aCSF; in mM: NaCl 120, KCl 2.8, Na$_2$HPO$_4$ 1, MgCl$_2$ 0.876, NaHCO$_3$ 22, CaCl$_2$ 1.45, glucose 2.55, pH = 7.4) was applied on the brain surface. An imaging coverslip (~4 × 4 mm, 0.08-mm thick; Menzel-Gläser) was positioned onto the craniotomy, leaving a ~0.5-mm gap for glass microelectrode insertion. The animal was transferred to the imaging stage, and the anesthesia was changed to continuous infusion of α-chloralose (50 mg/kg BW per hour) via an intravenous catheter.

Mice were allowed to rest for 25 min before the imaging procedures. Prior to imaging, a ~50 µL blood sample was collected via the arterial catheter for blood gas evaluation (ABL, Radiometer), and the respiration rate and volume were adjusted if necessary. To ensure physiological conditions, we monitored end-tidal CO$_2$ levels and MABP, and body temperature was maintained at 37 °C using a rectal thermistor-regulated heating pad.

**Animal preparation for chronic/awake imaging**. The surgery was performed as previously described, with minor modifications[73]. Briefly, 4 h prior to the surgery, the animals were injected with dexamethasone (4.8 mg/g BW; Dexavit, Vital Pharma Nordic). The anesthesia was induced with 5% isoflurane (ScanVet) in O$_2$-supplemented air (10%). Eyes were lubricated with eye ointment (Viscotears, Novartis), and the animal's head was shaved and mounted onto a stereotactic frame. The body temperature was maintained during all steps of the surgery at

37 °C using a rectal thermistor-regulated heating pad. Surgery was performed in an aseptic environment with heat-sterilized surgical tools.

The shaved skin was disinfected with chlorhexidine/alcohol (0.5%/74%; Kruuse). Next, carprofen (5 mg/kg BW; Norodyl, Norbrook), buprenorphine (0.05 mg/kg BW; Temgesic, Indivior), and lidocaine (100 µL 0.5%) were subcutaneously injected under the scalp. The anesthesia was reduced to 1.8–2.0% isoflurane, the scalp was removed, and the bone surface was cleaned from the periosteum with an ultrasonic drill (Piezosurgery, Mectron). A craniotomy was performed over the right somatosensory cortex (2 mm lateral, 0.5 mm posterior to bregma; Ø = 3 mm), the bone flap was carefully lifted, and the exposed brain temporarily covered with a hemostatic absorbable gelatin sponge (Spongostan®, Ferrosan, Denmark) pre-wetted with ice-cold aCSF. The cranial opening was filled with aCSF, then sealed with an autoclave-sterilized round imaging coverslip (Ø = 4 mm, 0.17-mm thick; Laser Micromachining LTD). The rim of the coverslip was secured with a thin layer of Vetbond (3M Company), and a lightweight stainless steel head plate (Neurotar) was positioned on the top of the skull in alignment with the cranial window. The skull was coated with adhesive resin cement (RelyX Ultimate, 3M Company) to secure the exposed bone, including the skin incision rim, and to firmly attach the metal plate to the head. Next, the animals were transferred onto a pre-warmed heating pad to wake from anesthesia (~5 min) in a cage supplemented with pre-wetted food pellets for easy chow and hydration.

Post-operation care consisted of subcutaneous injections of Temgesic (3 h) and Rimadyl (24 and 48 h post-surgery, doses as before). Animal welfare was closely monitored during the 7 days of post-surgical recovery and subsequent imaging training procedures. All animals underwent recurrent 30-min/day training sessions before the imaging to gradually habituate to the mobile cage system (Neurotar) with sugar-supplemented water as a reward (~14 days training). Given that no catheters were mounted in chronically imaged animals, the nanoparticles were injected retroorbitally 10 days post-surgery during brief (~2 min) isoflurane anesthesia (5%). This administration route was preferential to, e.g., tail vein injections, because it provided better control over the injectant volume. The imaging sessions never exceeded 45 min.

**Nanoparticle preparation**. 1,2-Distearoyl-sn-glycero-3-phosphocholine (DSPC), ovine cholesterol, 1,2-distearoyl-sn-glycero-3-phosphoethanolamine-N-[methoxy (polyethylene glycol)-2000] ammonium salt (DSPE-PEG2k), and 1,2-distearoyl-sn-glycero-3-phosphoethanolamine-N-[maleimide(polyethylene glycol)-2000] ammonium salt (DSPE-PEG2k-maleimide) were purchased from Avanti Polar Lipids (Alabaster). The stealth nanoparticles were prepared to consist of DSPC/cholesterol/DSPE-PEG2k (molar composition: 56.3:38.2:5.5), and the antibody-functionalized nanoparticles to consist of DSPC/cholesterol/DSPE-PEG2k/DSPE-PEG2k-maleimide (molar composition: 56.3:38.2:5:0.5). The fluorescent nanoparticles were supplemented with 0.5 mol% of 1,2-dipalmitoyl-sn-glycero-3-phosphoethanolamine labeled with Atto 488 (Atto 488 DPPE) or Atto 550 (Atto 550 DPPE; Atto-Tec). We chose DPPE-anchored Atto dyes because of their fairly low dissociation rate from liposome nanoparticles in plasma[74], and their high fluorescence quantum yield and photostability. To obtain lipid powder mixtures of the above compositions, the constituent lipids were dissolved in tert-butanol (Sigma-Aldrich)/Milli-Q water solution (9:1) and heated to 40–50 °C to ensure complete dissolution. The lipid solutions were then plunge-frozen in liquid N$_2$ and lyophilized overnight to remove the solvent (ScanVac CoolSafe lyophilizer, LaboGene).

**Nanoparticle fluorescent tagging**. To obtain fluorescently labeled nanoparticles, the lyophilized lipids were hydrated in 70 °C phosphate-buffered saline (PBS; 10 mM phosphate, 137 mM sodium chloride, 2.7 mM KCl, pH 7.4; tablets from Sigma-Aldrich) to a 50 mM lipid concentration. The lipid suspensions were vortexed every 5 min for a total period of 30 min, then subjected to five freeze-thaw cycles by alternate placement in a liquid N$_2$ and a 70 °C water bath. Next, the lipid suspensions were extruded 21 times through a 100-nm polycarbonate filter (Whatman, GE Healthcare) at 70 °C using a mini-extruder (Avanti Polar Lipids) to form nanoparticles.

**Nanoparticle targeting**. We used a high-affinity ($K_D$ = 6 nM) monoclonal anti-TfR antibody clone RI7217 to functionalize the nanoparticles[21]. The antibody was produced in-house using the hybridoma technique at Laboratory for Neurobiology, Aalborg University, Denmark. The antibody specificity was previously determined using surface plasmon resonance[21]. To functionalize the nanoparticles with either RI7217 or a rat isotype IgG control (Thermo Fisher Scientific, Waltham, MA, USA), we prepared solutions of 8 mg/mL antibody in borate buffer (100 mM borate, 2 mM EDTA, pH 8.0; all Sigma-Aldrich). The antibody concentrations were determined from the absorbance at 280 nm (NanoDrop 2000c spectrophotometer, NanoDrop Products, Thermo Fisher Scientific) using mass extinction coefficients of 1.3 (mg/mL)$^{-1}$ cm$^{-1}$ and 1.5 (mg/mL)$^{-1}$ cm$^{-1}$ for RI7217 and IgG isotype, respectively, determined in a separate micro-BCA experiment. Traut's reagent (Thermo Fisher Scientific) was added to a reagent-to-antibody molar ratio 10:1 in Protein LoBind tubes (Eppendorf), and the solutions were incubated for 1 h at room temperature under constant shaking at 500 rpm. Using Amicon Ultra-4 30 kDa centrifugal filter units (Merck), we transferred the thiolated antibodies to PBS and determined their concentration using the NanoDrop 2000c as described above.

Next, we added 1.05 mg of newly prepared thiolated antibodies to 700 μL of newly prepared nanoparticles (lipid concentration ~35–40 mM) in Protein LoBind tubes and replaced the air phase in the tubes with N$_2$. The samples were then incubated for 24 h at room temperature under constant shaking at 500 rpm, allowing the thiolated antibodies to couple to the maleimide groups on the surface of the nanoparticles. The antibody-functionalized nanoparticles were separated from unbound antibodies using a Sepharose CL-4B (GE Healthcare) size-exclusion chromatography column eluted with PBS (dimensions, 1.5 × 20 cm; flow rate, 1 mL/min). The recovered nanoparticles were pooled in Amicon Ultra-4 100 kDa centrifugal filter units (Merck) and concentrated by centrifuging at 2000 × g until the lipid concentration was increased to 30–40 mM.

**Nanoparticle cisplatin loading, targeting, and detection.** To prepare cisplatin-loaded nanoparticles, *cis*-diammineplatinum(II) dichloride (cisplatin; Sigma-Aldrich) was dissolved in PBS to a nominal concentration of 8.5 mg/mL. The solutions were magnetically stirred for 1 h at 70 °C and subsequently left at room temperature for 15 min, allowing undissolved cisplatin crystals to precipitate. The supernatants were transferred to new vials and magnetically stirred while being heated to 70 °C. The solutions were then added to lyophilized lipids, resulting in 50 mM lipid suspensions that were magnetically stirred for 1 h at 70 °C and extruded as described above for the fluorescently labeled nanoparticles. The samples were cooled to room temperature to allow any residual cisplatin crystals to precipitate, and the supernatants were run on a Sepharose CL-4B size-exclusion chromatography column eluted with PBS (dimensions 1.5 × 20 cm, flow rate 1 mL/min) to remove free cisplatin. The recovered nanoparticles were concentrated using Amicon Ultra-4 100 kDa centrifugal filter units by centrifuging at 2000 × g.

To prepare antibody-functionalized cisplatin-loaded nanoparticles, the antibodies were thiolated as described above. Then, 0.5 mg of newly prepared thiolated antibody was added to 700 μL of newly prepared cisplatin-loaded nanoparticles (lipid concentration ~13 mM) in a Protein LoBind tube. The nanoparticles were then incubated, recovered, and concentrated as described above for the fluorescently labeled antibody-functionalized nanoparticles.

Brain uptake of cisplatin was measured with inductively coupled plasma mass spectrometry (ICP-MS) as recently described[22]. Briefly, a maximum of 100 mg brain tissue input was digested in 65 °C aqua regia overnight, then diluted in 0.5 ppb iridium aqueous solution (Sigma-Aldrich). Next, the samples were diluted in 2% HCl, 0.5 ppb iridium aqueous solution, and immediately analyzed using an iCAP Q ICP-MS system (Thermo Scientific) equipped with a Cetac ASX-520 AutoSampler and a Neslab ThermoFlex 2500 chiller. Measurements were performed in standard mode with iridium as an internal standard, and platinum levels were determined by comparing to a platinum standard curve in the range 0.08–5 ppb (Sigma-Aldrich). The data were represented as the percent of injected dose per gram of the brain tissue (%ID/g).

**Nanoparticle properties in vitro.** The phosphorus concentrations of the nanoparticle stock samples were determined using ICP-MS. The samples were diluted in 2% HCl, 10 ppb gallium aqueous solution (Sigma-Aldrich), and the measurements were done in kinetic energy discrimination mode with gallium as an internal standard. Phosphorus levels were determined by comparing to a phosphorus standard curve in the range 25–100 ppb (Sigma-Aldrich). The phospholipid concentrations were estimated by subtracting the phosphorus background of the PBS buffer, and the total lipid concentrations estimated by dividing the phospholipid concentrations with 0.618, taking into account that cholesterol does not contain phosphorus. For the cisplatin-loaded nanoparticles, we also used ICP-MS to determine the platinum concentrations. The samples were diluted in 2% HCl, 0.5 ppb iridium aqueous solution, and the measurements were performed in standard mode with iridium as an internal standard. Platinum levels were determined by comparing to a platinum standard curve in the range 0.125–1 ppb. The size of the nanoparticles (dissolved in PBS) was measured using dynamic light scattering, and the zeta potential of the nanoparticles in phosphate-glucose buffer (10 mM phosphate, 280 mM glucose, pH 7.4; reagents from Sigma-Aldrich) was measured using mixed measurement mode phase analysis light scattering (Zetasizer Nano ZS, Malvern Instruments). The antibody conjugation level on the functionalized nanoparticle was determined using the micro-BCA assay (reagents purchased from Thermo Fisher Scientific), performed by incubating samples (including bovine serum albumin [BSA] standard samples) for 1 h in a 60 °C water bath and then transferring them to a 96-well plate to measure their absorbance at 562 nm using a Spark multimode microplate reader (Tecan). To account for the small contribution of lipids in the micro-BCA assay[74], we also performed the micro-BCA on non-functionalized nanoparticles, which allowed for the subtraction of the lipid contribution to determine the amount of antibody conjugated to the nanoparticles. The hydrodynamic diameter ($D_h$) of RI7-functionalized Atto 550-tagged (RI7-L-A550) and Atto 488-tagged (RI7-L-A488) nanoparticles was in the range of $D_h$ = ~135–140 nm (Supplementary Table 1) and comparable to other TfR-targeted clinically relevant formulations[57,67]. Both RI7-L-A550 and RI7-L-A488 had a low polydispersity index (≤0.13), indicating high size homogeneity. Assuming the nanoparticles contained on average ~2.5 × 10$^5$ lipids, the conjugation level of 30 g/mol$_{lipid}$ corresponded to ~50 antibodies per nanoparticle[74].

**Fluorescent probes.** FITC-dextran (MW 2 MDa, 0.5%, Sigma-Aldrich), TRITC-dextran (MW 65 kDa, 1%, Sigma-Aldrich), or bovine serum albumin Alexa Fluor 488-conjugate (BSA-Alexa 488, 1%, Invitrogen) was administered as a single bolus injection (50 μL) via a femoral arterial catheter. All were dissolved in sterile saline and administered subsequently to nanoparticles. In addition to delineating a vessel lumen, lack of extravascular leak of dyes indicated preserved BBB structural integrity after the microsurgery.

**Imaging setup.** In vivo two-photon imaging was performed with an SP5 upright laser scanning microscope (Leica Microsystems) coupled to MaiTai Ti:Sapphire laser (Spectra-Physics). The images were collected using a 20× 1.0 NA water-immersion objective. The fluorescence signal was split by FITC/TRITC filter and collected by two separate multi-alkali photomultipliers after 525–560 nm and 560–625 nm bandpass filter (Leica Microsystems). The fluorophores were excited at 870 nm with the 14 mWatt output power at the sample. The images were collected using LAS AF v. 4.4 (Leica Microsystems) in 16-bit color depth and exported to ImageJ for further analysis (v. 1.52a; NIH). 3D reconstructions were performed via volume rendering in Amira v. 6 (FEI Visualization Sciences Group).

**Surface density calculation.** To assess spatio-temporal properties of nanoparticles association to the endothelium, we monitored the association of nanoparticles for 2 h after injection with respect to all cerebral vessel types using hyperstack (4-dimensional) imaging. Data were recorded as a series of Z-stacks in bidirectional scanning mode with triple frame averaging, from 387.5 μm × 387.5 μm area (2048 × 2048 pixel resolution) and 144 μm depth span (Z-step size 2.50 μm) with 7.5-min intervals between consecutive Z-stacks. The nanoparticles were counted from all vessels in the field of view, with each individual vessel followed over time. The vessel surface area was calculated from vessel diameter delineated by FITC-dx or TRITC-dx signal and the length of the vessel measured in 3D. The association density was obtained from a nanoparticle count per corresponding vessel wall area [nanoparticles/μm$^2$].

**Nanoparticle uptake estimation.** The precise spatial localization of nanoparticles in microscopy is non-trivial because of the diffraction limit; therefore, we defined the boundary conditions characterizing nanoparticle location using the previously calculated nanoparticle fluorescence signal spread, i.e., standard deviation ($\sigma$) of Gaussian fluorescence distribution ($\sigma = 0.29$ μm; Fig. 1j). The nanoparticles were categorized as internalized only when separated from endothelium peak signal by |$\Delta p$| < $\sigma$, and as adhering (adh.) when separated from the endothelium peak signal by at least 2$\sigma$, i.e., $\Delta p > 2\sigma$. Additional groups consisted of nanoparticles found on the abluminal side (abl.), i.e., when $\Delta p < -2\sigma$; and nanoparticles that could not be categorically classified into any of the groups above and represented the intermediate groups, i.e., unresolved on luminal side (unL) when $\sigma < \Delta p < 2\sigma$; and unresolved on the abluminal side (unA) when $-2\sigma < \Delta p < -\sigma$ (Fig. 4f).

**Subcellular distribution mapping.** We imaged the surface of the vessels, i.e., a ~5 μm planar optical section aligned with the vessel circumference at 3 h post-injection. When measuring distances from the nucleus geometric center to nanoparticles (ImageJ), we set the distance cut-off point to 11 μm to exclude the nanoparticles that belonged to neighboring endothelial cells and to avoid distribution bias due to the non-concentric spindle-like geometry of endothelial cells. In addition, we excluded nanoparticles located in line from the geometric center toward the vessel wall, where the cut-off distance exceeded the vessel boundary. We took this step to minimize the effect of the vessel curvature on the estimation of the distance.

**Nanoparticle tracing.** Nanoparticles were manually tracked (ImageJ) 2–3 h post-injection throughout 30 min of continuous imaging ($\Delta t$ between frames = 30 s) at the vessels with the long symmetry axis aligned with the imaging plane. To avoid systemic differences in nanoparticle tracing experiments, we analyzed vessels aligned with the focal plane, i.e., oriented perpendicular to the imaging axis. Tracing nanoparticles from penetrating vessels that traversed the imaging plane under a steep angle, e.g., ascending venules, might underestimate the movement along the long vessel symmetry axis *v*. Furthermore, the nanoparticles would only briefly appear in the imaging plane, and in order to obtain longer time recordings (30 min), the imaging would be limited to nanoparticles that exhibited only a small degree of movement to remain in the focal plane.

**MSD analysis.** To characterize the motion of nanoparticles, we used the mean square displacement (MSD) analysis[75]. Time-lapse recordings were collected in bidirectional scanning mode from 387.5 μm × 387.5 μm area (2048 × 2048 pixel resolution) for 30 min at 30-s intervals (60 data timepoints). Each nanoparticle trajectory was manually traced, treating the centroid of nanoparticle 2D fluorescence intensity as a location coordinate. The planar ($x,y$) trajectories were projected to a vector **V** aligned with the vessel long symmetry axis *v* and with the direction of the blood flow. For every trajectory, the displacements in *v* coordinate in time *t* were extracted for each multiplier of the smallest resolved time interval *d* (i.e., *t* = *d*, *t* = 2*d*, *t* = 3*d*…*t* = *i\*d*), where *i* = 60 timepoints and *d* = 30 s. The

displacements were squared and averaged between nanoparticles for each respective time interval $t = i*d$. The significant deviation of MSD($\Delta t$) from linearity with the increase of $t$ indicates a non-stochastic (directional) movement component[75]. We assessed MSDv($\Delta t$) linearity with the least-squares linear regression fit weighted by the inverse of data point variance, followed by Wald–Wolfowitz runs test.

**Electrophysiological recordings.** Electrocortical brain activity (ECoG) was recorded via a heat-pulled borosilicate glass electrode containing an Ag/AgCl filament and filled with aCSF (electrode tip Ø, 2–3 μm; inner Ø, 0.86 mm; outer Ø, 1.5 mm; Sutter Instrument; resistance 1.5–2.0 MΩ). The electrode was inserted under the glass coverslip ~50 μm into the cerebral cortex, and the reference electrode was positioned in the neck muscle. The total electrical signal was filtered (3000 Hz low-pass filter), then amplified 10× (AP311 analog amplifier; Warner Instruments), and the alternate current-ECoG component (i.e., spontaneous brain activity) was obtained after further 100× amplification and 0.5 Hz high-pass filter (NL 106 analog amplifier and NL 125/126 analog filter, NeuroLog). Analog data were digitized (Power 1401, CED) at 20 kHz. For the exhaled $CO_2$, MABP (the raw readout) was collected. All data were recorded in Spike2 software (v. 7.02a; CED).

**Statistics and reproducibility.** Following the D'Agostino-Pearson normality test, an unpaired two-tailed Student's $t$-test or two-tailed Mann–Whitney tests were used for data with normal or non-normal distribution, respectively, unless stated otherwise, i.e., in Fig. 5f. For multiple comparisons between groups, Bonferroni post hoc corrections were used. The number of analyzed nanoparticles or vessel segments across the biologically independent mice is provided on figure legends and on figures with the number of mice in brackets. All statistical analyses were performed in Prism v.8.2 (GraphPad). Data were plotted in Prism v.8.2. or in OriginPro 2018 (OriginLab Corporation). The panels with representative images were selected from $n$ mice showing similar results; where: Fig. 1f $n_{RI7-L-A550} = 4$, $n_{RI7-L-A488} = 4$; Fig. 1h $n_{RI7-L-A550} = 3$, $n_{RI7-L-A488} = 3$; Fig. 1i $n = 1$; Fig. 2a–f $n = 7$; Fig. 2g, h $n = 1$; Fig. 3a, b $n = 4$; Fig. 3d $n = 4$; Fig. 4a–e $n = 5$; Fig. 5a, b $n = 5$; Fig. 5g–i, l $n = 5$; Fig. 6a $n_{upper\_panel} = 4$, $n_{lower\_panel} = 3$; Fig. 6b $n = 3$; Fig. 6c $n = 2$; Fig. 6d $n = 4$; Fig. 6g, h $n = 2$; Supplementary Figs. 1a, c $n_{RI7-L-A550} = 6$, $n_{RI7-L-A488} = 5$; Supplementary Fig. 1d $n = 2$; Supplementary Fig. 2a–d $n = 2$; Supplementary Fig. 3a $n = 5$. Supplementary Fig. 3b–e $n = 2$. Images from single experiments were included to illustrate the lack of targeting properties of nanoparticles without TfR-targeting antibody, i.e., in Supplementary Fig. 1f, g $n = 1$; Supplementary Fig. 1h, i $n = 1$; Supplementary Fig. 2e $n = 1$; Supplementary Fig. 4 $n = 1$ mice.

**Exclusion criteria.** Prior to injection of nanoparticles, all animals with abnormal blood pressure (<50 mmHg), abnormal brain ECoG activity, or significant (>2 μm/min) brain movement in x, y, or z coordinates were excluded from the study (3 excluded out of total 36 animals for in vivo imaging). No animals were excluded from immunohistochemistry analysis (total = 2 mice injected with RI7-L-A550 for confocal imaging) and from ICP-MS (total = 13 mice).

**Reporting summary.** Further information on research design is available in the Nature Research Reporting Summary linked to this article.

## Data availability
The data that support the findings of this study are available within the article and its Supplementary Information files or from the corresponding author upon reasonable request. Source data are provided with this paper.

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

## Acknowledgements

This study was supported by Lundbeck Foundation Research Initiative on Brain Barriers and Drug Delivery (#R392-2018-2266); Det Frie Forskningsråd (#0602-01965B); and Nordea Foundation Grant to the Center for Healthy Aging (#114995). We thank Nikolay Kutuzov for feedback on data analysis, Serhii Kostrikov for his assistance in cisplatin experiments, and Kjeld Møllgård for his comments on the manuscript.

## Author contributions

Conceptualization, Kr.Ku., Ka.Kr., and M.J.L.; methodology, Kr.Ku., Ka.Kr., and K.B.J.; formal analysis, Kr. Ku.; investigation, Kr.Ku., Ka.Kr., K.B.J., M.A.L., and M.L.; resources, T.M., T.L.A., and M.J.L.; writing—original draft, Kr.Ku., and M.J.L.; writing—review & editing, Kr.Ku., Ka.Kr., K.B.J., T.M., and M.J.L.; visualization, Kr.Ku.; funding acquisition, T.M., T.L.A., and M.J.L.; supervision, M.J.L.

## Competing interests

The authors declare no competing interests.
