## [Peer Review File · Nature Communications]

Reviewers' Comments:

Reviewer #1:

Remarks to the Author:

Kucharz, et al. are investigating the fate of targeted nanoparticles for delivery of therapeutic cargo to the brain parenchyma. Using two photon microscopy, they show that liposomes functionalized with an anti-transferrin receptor antibody bind primarily to capillaries and venules in the brain. Over time, they find particles bound to the veins/venules can pass through to the abluminal side of the vessels. Interestingly, they find in venules, internalized nanoparticles traffic to perinuclear areas.

The tracking of single nanoparticles over time is impressive and some of the differences across vessel segments (from arteries through to veins) are quite striking. As transferrin receptor mediated BBB crossing is a leading possibility for therapeutics, an understanding of how this is being mediated is important. However, there are a few important issues arising in the manuscript. Major points

1. Mechanism of BBB-crossing is unexplored

Starting in the abstract the authors claim that transit across the endothelium takes place in venules because of greater perivascular space. However, the experimental evidence they present for this hypothesis is sorely limited. For delivery of a molecule through the transcellular pathway from the blood to the brain parenchyma, the molecule must (a) endocytose at the luminal membrane of the endothelial cell, (b) exocytose at the abluminal membrane, and (c) transit further into the parenchyma. The authors do show convincing images of nanoparticles effectively getting out into the parenchyma around venules while not around capillaries (figure 6) – this is good evidence for part (c) of this pathway. However, the spatial resolution of the experiment does not allow for convincing evidence that there is no transcytosis.

To this end, the subcellular distribution of nanoparticles presented in figure 5 is notable. Although the authors claim in the discussion that this perinuclear distribution corresponds to late endosomes, this is by no means the only possibility. Indeed, the common endosome (recycling endosome) is specifically thought to transit through this perinuclear space for transcytosis (see, for instance, Apodaca et al1)! This would suggest a meaningful intracellular trafficking difference between cECs and vECs, not necessarily having anything to do with perivascular space.

The laser extravasation results in figure 6 point to some kind of difference between capillaries and veins in regards to efficacy of nanoparticle drug carriers. But by no means do these results prove that the difference in perivascular space is the reason.

The distinction between access to the parenchyma and transcytosis is an important one. It would be best to stick with the verbiage of the heading on page 9 – namely “impedes nanoparticle brain entry,” as that is what the results show.

2. The authors present this as an important avenue for therapeutics, but their results point to this being an ineffective mechanism.

Most approaches for delivering therapeutic molecules across the BBB tend to target the capillaries. This is because capillaries are by far the most abundant vascular segment. By contrast, veins/venules are the least abundant. Their data also suggests that few particles accumulate further into the parenchyma and are instead stuck in the perivascular space even after several days. How does that affect therapeutic potential? Could this persistent accumulation lead to adverse effects, even? The authors only show data claiming no adverse effects during a two hour imaging window (figure S1). But in some of their previous work with the same nanoparticle/antibody preparation they actually found significant adverse effects².

Given that therapeutic benefit would likely come from repeated or extended exposures, time points beyond 2 hours should be explored to address this idea of a lack of adverse effects. Furthermore, in line with this, it is becoming increasingly clear that post-capillary venules and the perivascular spaces are immunologically active under homeostatic conditions. Thus, it would therefore be warranted to ensure that nanoparticle delivery across this vascular segments and accumulation into these spaces does not disrupt homeostatic immune cell transmigration, which could potentially yield devastating side effects.

3. Claims based on two photon imaging would benefit from validation by histology

The authors rightly point out that tissue processing for histology can affect measurements of luminal bound nanoparticles. However, perfusion with fixative should effectively preserve any

particles that are truly endocytosed and have no effect on their distribution (nor on the distribution of particles that have transited across the BBB completely). The live imaging is very important for this paper, but follow up by histology would provide the following:

a) Superior spatial resolution. This would allow for more rigorous discrimination between particles that are truly inside versus outside endothelial cells.

b) Greater understanding of molecular differences. For instance, transitions between capillaries and veins are not clearly identifiable by morphology and these areas could be potentially interesting. Indeed, a panel of carefully identified markers used in combination would likely be necessary to rigorously identify specific vascular segments, a strategy used in this study.

c) Broader sampling throughout other areas of the brain, including various clinically interesting areas not in the somatosensory cortex. Specifically, if nanoparticle delivery is primarily effective around only a subset of vessel segments it would be interesting to understand how those are distributed in the brain.

This would also alleviate any concerns that imaging in an area where there was just surgery performed may affect the distribution of nanoparticles.

Minor points

- Flow through different vessel segments varies significantly (velocity, turbulence, etc). It would be interesting to include some discussion how flow might play into the distribution of binding they observe
- Is the MSD analysis presented in figure 5 from all vessel segments observed or specifically from veins?
- At what time point from nanoparticle delivery is the subcellular distribution in 5j from?
- Numbers of mice and numbers of individual vessels should be indicated more clearly (in addition to numbers of particles). For instance, it is not clear how many mice were used for the experiment in figure 6.

1. Apodaca, G., Katz, L. A. & Mostov, K. E. Receptor-mediated transcytosis of IgA in MDCK cells is via apical recycling endosomes. *J. Cell Biol.* 125, 67–86 (1994).

2. Johnsen, K. B. et al. Modulating the antibody density changes the uptake and transport at the blood-brain barrier of both transferrin receptor-targeted gold nanoparticles and liposomal cargo. *J. Control. Release* 295, 237–249 (2019).

Reviewer #2:

Remarks to the Author:

I find the work from Kucharz et al. exceptionally well executed and with findings that can have a potential impact on several communities, including vasculature biology, blood-brain barrier physiology, nanomedicine, and neurology. The quality of the in vivo imaging is supreme and equally matched by elegant and thorough data analysis. Such efforts alone make the work suitable for publication in ANY journal including Nature Communications.

However, before I can recommend the work to be published, I feel the authors should address two critical aspects:

1) Imaging depth:

I don't have a clear sense of where most of the imaging is executed and perhaps a detailed explanation either in the form of more information in the method or by making the scheme in figure 1b larger indicating in which part of the brain where most of the imaging takes place. It appears the most blood vessels are part of the pia mater and hence while technically considered part of the brain vasculature, their location within the meninges make them potentially phenotypically different from the brain parenchyma vasculature. They are characterised by less belong to an anatomical area which is lower pericyte density, served by a functioning lymphatic system and is connected to the peripheral nervous system via extrinsic innervations. All of these are the characteristics of a non-cerebral vasculature. While I can see why the author focussed on more accessible pial vessels, I wonder how much information is accessible from parenchymal vasculature and their implication in the whole blood-brain barrier.

2) Tf affinity/avidity

I'm not sure why the authors chose to use a high-affinity antibody against TfR, which is even more augmented by the liposome multivalency. It is now well established that ligands with high affinity (ref 33 in the manuscript) or high avidity (Ref 27, Niewoehner et al. Neuron. 2014; 81: 49-60) for the transferrin receptor (TfR) cannot cross the BBB but instead redirect receptors to lysosomes for degradation. Villaseñor et al. (10.1016/j.celrep.2017.11.055) showed that such an effect is regulated at the endosomes sorting level. More recently, we established a similar impact of avidity to LRP1 receptors in modulating transcytosis (10.1126/sciadv.abc4397) as well as showing two distinct pathways of intracellular transport, one associated with the F-bar protein syndapin-2 that leads to fast transcytosis with no endosomal sorting, and one that leads to endothelial internalisation and associated with endosomes and lysosomes.

The liposomes chosen here belong to the high avidity class. The data show very eloquently their ability to enter endothelial cells and guide the TfRs toward lysosomes in agreement with the existing literature. All the discussion about deficient transcytosis and its relation with the venules/capillary etc. becomes redundant. While I would love to see a similar analysis repeated with nanoparticles whose avidity is adequately optimised for transcytosis, I can see that such a request involves a considerable amount of work. On the other hand, I repeat myself, and I think the methodology here presented is very relevant and warrants publication. Perhaps a rewriting of the discussion with consequent moderation of the current conclusions to include the current state-of-the-art on transcytosis might be still sufficient to justify the presented work keeping it compelling. One example is the role of perivascular space, and the recent conclusion is supported only if the TfR targeting liposomes were able to go via transcytosis.

Minor comment:

General: I find the manuscript quite challenging to read, perhaps too segmented, with little flow between one section to the other. The figure captions are not useful and too hermetic.

Page 3, Line 5: what's the difference between low molecular weight hydrophobic compounds and lipophilic drugs? Are the latter part of the former? Perhaps this sentence needs some clarification.

Page 6, line 13, how do the authors ensure that the overlapping signal is not an aggregation or an exchange between the two liposome formulations? Let's remind ourselves that the components that make liposomes have a CMC>0 and thus exchange via assembly disassembly processes.

Page 8, line 6: Can the authors explain the omission with more details?

Page 9, line 8-11. Is it possible that this is due to the advection within the perivascular space (see the work from Nedergaard and Iliff on the glymphatic system)

Prof Giuseppe Battaglia
Catalan Institution for Research and Advanced Studies
Institute for Bioengineering of Catalonia - Barcelona Institute of Science and Technology
Department of Chemistry and Institute for the Physics of Living Systems -University
College London

Reviewer #3:

Remarks to the Author:

The blood brain barrier (BBB) is critical for maintaining the health of the brain, but is a major impediment to delivering drugs and other substances in a targeted manner. In order to get the substance of interest across the BBB, nanoparticles with surface moieties that make use of the endogenous transport machinery (usually transferrin) have been developed. However, where in

the vasculature this transport is taking place was not known, as most of the studies had looked histologically, or with low resolution techniques like MRI. In this manuscript, the authors use in vivo two-photon microscopy to visualize where in the vascular network transport takes place. They find while a large number of particles are taken up in the capillary bed, the transport of the particles stall within the endothelial cells of capillaries and thus do not make it into the brain. They find that most of the trans-vascular transport takes place in the venules, which they hypothesize to be due the larger perivascular space around venules. I think this is a very nice manuscript. In general, the data is very clear and convincing. The analyses are well done and there is a great deal of attention paid to the limitations of the optical resolution in resolving the nano particle positions. Although the paper is largely observational, I think it is an important advance to the field and will guide the development of more effective BBB-crossing drug delivery systems.

I have only minor comments, listed below.

The cited reference for transferrin expression in the capillaries is nearly four decades old, and the technology to assay single-cell level expression differences has greatly improved since then. What do more recent measurements of single-cell gene expression (such as those from the Betsholtz lab and others) show with regards to the expression patterns of transferrin (and other proteins involved in transcytosis) in the cerebral vasculature?

Why do capillaries have the machinery for uptake but not complete transcytosis? This is odd. Is this a consequence of the size of the particles, and smaller particles might be transcytoses at a higher rate?

The larger perivascular space around venules seems to be a plausible explanation for the observed results, but in fairness, the authors do not conclusively rule out other hypotheses. Laser extravasation may produce different opening magnitudes and flows of fluid out of venules and capillaries, so this experiment is suggestive, but not conclusive. I think in light of this, the conclusion that the perivascular space difference could be toned down a bit.

REVIEWER #1

Reviewer Comments:

Reviewer #1: Kucharz, et al. are investigating the fate of targeted nanoparticles for delivery of therapeutic cargo to the brain parenchyma. Using two photon microscopy, they show that liposomes functionalized with an anti-transferrin receptor antibody bind primarily to capillaries and venules in the brain. Over time, they find particles bound to the veins/venules can pass through to the abluminal side of the vessels. Interestingly, they find in venules, internalized nanoparticles traffic to perinuclear areas. The tracking of single nanoparticles over time is impressive and some of the differences across vessel segments (from arteries through to veins) are quite striking. As transferrin receptor mediated BBB crossing is a leading possibility for therapeutics, an understanding of how this is being mediated is important. However, there are a few important issues arising in the manuscript.

We thank Reviewer #1 for his/her appreciation of our work and insightful comments.

Major points

Rev1_Comment #1a. Mechanism of BBB-crossing is unexplored. Starting in the abstract the authors claim that transit across the endothelium takes place in venules because of greater perivascular space. However, the experimental evidence they present for this hypothesis is sorely limited. For delivery of a molecule through the transcellular pathway from the blood to the brain parenchyma, the molecule must (a) endocytose at the luminal membrane of the endothelial cell, (b) exocytose at the abluminal membrane, and (c) transit further into the parenchyma.

We agree with Reviewer #1 that transcytosis is a multiple-step process, and the distinction between transcytosis across endothelium (with successful exocytosis) and transcytosis followed by progression to the parenchyma is an important one. Here, we used the term transcytosis when referring to successful nanoparticle entry into the brain, which also indicated that the transcytosis was complete, i.e., the event of exocytosis from endothelium must have occurred.

We thank Reviewer #1 for this comment, and to clarify, we have now revised the text using the term “transcytosis” to relate to the steps (a)+(b); and “transcytosis-mediated brain entry/delivery/transit” to relate to the steps (a)+(b)+(c).

Rev1_Comment #1b. The authors do show convincing images of nanoparticles effectively getting out into the parenchyma around venules while not around capillaries (figure 6) – this is good evidence for part (c) of this pathway. However, the spatial resolution of the experiment does not allow for convincing evidence that there is no transcytosis.

We agree with Reviewer #1 concerning the limitations of the imaging resolution but would like to point out that we don't exclude the possibility of nanoparticle transcytosis in capillaries. However, our data generally suggest that the incidence of nanoparticle transcytosis at capillaries had to be low compared to post-capillary venules. In particular, despite the highest binding of nanoparticles to capillaries (Manuscript Fig. 3c), and high retention of nanoparticles in the brain, as observed at post-capillary venules (Manuscript Fig. 6g, h, j), there was no significant nanoparticle presence at the capillary walls at 24 and 48 hours post-injection (Manuscript Fig. 6g, j). Had the transcytosis occurred to a similar degree as in post-capillary venules, we would expect at least an equal amount of nanoparticle signal to be present at the capillary walls, regardless of whether we would be able to distinguish between the cellular or extracellular side of the endothelial cell.

We have now revised the Discussion and added the abovementioned argument to support the claim on the negligible contribution of capillaries to the total transport of nanoparticles to the brain [Page 13, Line 22].

Rev1_Comment #2a. To this end, the subcellular distribution of nanoparticles presented in figure 5 is notable. Although the authors claim in the Discussion that this perinuclear distribution corresponds to late endosomes, this is by no means the only possibility. Indeed, the common endosome (recycling endosome) is specifically thought to transit through this perinuclear space for transcytosis (see, for instance (Apodaca et al., 1994)!

Intracellular transport vesicles exhibit a characteristic pattern of distribution within brain endothelial cells observed at their flat projections (Toth et al., 2020; Toth et al., 2018). We wrote: "(...) Notably, in contrast to capillaries, post-capillary and pial venules exhibited the preferential distribution of nanoparticles to perinuclear areas, which corresponds to the position of late endosomes that contain components directed for degradation in the transcytotic pathway. (...)". This is in agreement with the perinuclear location of both late endosomes (Huotari and Helenius, 2011; Toth et al., 2019; Toth et al., 2018) and lysosomes downstream in the trafficking route (Toth et al., 2019; Toth et al., 2018). The perinuclear distribution of nanoparticles suggested that endocytosed nanoparticles converge with the lysosomal degradation pathway, similarly to previously observed distribution for high-affinity antibodies targeting TfR (Bien-Ly et al., 2014; Johnsen and Moos, 2016). However, as Reviewer #1 noted, it does not exclude the occurrence of nanoparticles in other compartments of intracellular transport, e.g., the contribution of recycling endosomal pathway. Noteworthy, the latter is typically distributed in proximity to the microtubule organizing center, localized at one spot close to the nucleus membrane (Toth et al., 2020; Toth et al., 2018).

We thank Reviewer #1 for the reference, which we have read with great interest. To relate our findings to Apodaca et al., or others, one must consider the potential impact of the fundamental functional and morphological differences between brain endothelium and epithelial cells (Abbott et al., 2010), such as the kidney cell line used in the referred work (Apodaca et al., 1994). The brain endothelial cells are thin, i.e., up to ~0.4 μm at processes, and ~2 μm where the cell volume is occupied mainly by the nucleus; compared to the much taller and larger intestinal (~7.5 μm) or kidney (~17 μm) epithelial cells (Hellinger et al., 2012). Given the minimal diameter of vesicles in the endo-lysosomal system ranging between 0.2 to 0.25 μm (Toth et al., 2020), the thinness of brain endothelial cells may effectively limit apical-to-basal polarity seen in other tissues. Moreover, compared to other cell types, under normophysiological conditions the transcytosis in the BECs is strongly suppressed (Armulik et al., 2010; Daneman and Prat, 2015), which further sets them apart from peripheral barriers.

Overall, we agree that the perinuclear distribution is not an indication of sorting only into late endosomes, and we have now clarified this in the manuscript [Page 15, Line 1]. However, we would like to stress that general assumptions from polarized epithelial cells in peripheral organs should be treated with caution when applying to functionally and morphologically distinct brain endothelium.

Rev1_Comment #2b. This would suggest a meaningful intracellular trafficking difference between cECs and vECs, not necessarily having anything to do with perivascular space. The laser extravasation results in figure 6 point to some kind of difference between capillaries and veins in regards to efficacy of nanoparticle drug carriers. But by no means do these results prove that the difference in perivascular space is the reason. The distinction between access to the parenchyma and transcytosis is an important one. It would be best to stick with the verbiage of the heading on page 9 – namely “impedes nanoparticle brain entry,” as that is what the results show.

We agree with Reviewer #1 that the distinct intracellular distribution of nanoparticles between venules and capillaries likely reflects the difference in intracellular sorting, and we don't consider that it is a result of the presence/absence of the perivascular space. As Reviewer #1 pointed out in Rev1_Comment #1a, these observations concern two separate aspects of the nanoparticle transfer to the brain.

Concerning transcytosis: until recently, the understanding of mechanisms of endothelial transport came primarily from in vitro BBB culture models, which do not preserve vascular segment heterogeneity and do not reflect well the molecular landscape of the endothelium in vivo (Sabbagh and Nathans, 2020). It is only recently when the possibility of vascular zonation was explored at the level of receptor proteins (Yang et al., 2020), transcriptome (Kalucka et al., 2020; Sabbagh et al., 2018; Vanlandewijck et al., 2018), and transport modulation (Janiurek et al., 2019). The difference in trafficking between capillary and venular endothelial cells observed herein may be closely associated with normal vessel function. It

is well established that transendothelial transport of endogenous protein receptor ligands (or antibody-based medicines) occurs at the capillary level (Abbott et al., 2010; De Bock et al., 2016). Conversely, post-capillary venules mediate transendothelial transport of immune cells surveilling the brain (Engelhardt et al., 2017). One can speculate that venular endothelial cells may contain intracellular machinery that can better handle large entities, like the nanoparticles used in our study. While we cannot ascertain whether TfRs on venules elicit similar cellular responses as adhesion molecules for immune cells, it is an exciting subject to explore in future studies.

Concerning perivascular space: in turn, after successful transcytosis, the lack of perivascular space may be a limiting factor for further brain entry of nanoparticles. Our laser-extravasation data demonstrates that the nanoparticles in the perivascular space at venules can move relatively freely, as opposed to the brain parenchyma at capillaries. A similar movement in perivascular spaces was also observed after the infusion of nanoparticles into the brain, suggesting that this is the route of the least resistance (Mestre et al., 2018), which facilitates the distribution of therapeutics in the brain (Hadaczek et al., 2006). This might make it easier for the nanoparticles to leave the basement membrane in post-capillary venules. In contrast, capillaries are not surrounded by perivascular space, thereby creating a higher resistance environment, which may restrict brain entry of the nanoparticles (Engelhardt et al., 2017; Owens et al., 2008). This is consistent with previous ex-vivo findings of nanoparticles being restricted from progressing further into the brain at capillary segments of the brain vascular network (Cabezon et al., 2015; Muldoon et al., 1999).

These two mechanisms are not mutually exclusive, and we concluded that both intracellular sorting and perivascular space presence independently influence nanoparticle delivery to the brain.

We have now revised the Discussion with the information above [Page 13, Line 22; Page 14, Line 11]. In addition, we have now corrected the verbiage to reflect the essence of the subheadings more clearly.

Rev1_Comment #3a. The authors present this as an important avenue for therapeutics, but their results point to this being an ineffective mechanism.

The transcytosis and subsequent brain entry of nanoparticles may occur to a less degree than free antibodies, but this is compensated by a large amount and various types of drugs that can be encapsulated in a liposome nanoparticle (Huwlyer et al., 1996), and liposomes mono-targeted with antibodies to TfR outperform other receptor-mediated delivery targets in the brain in vivo (van Rooy et al., 2011). We do not claim that this is good in all its aspects, and in fact, we point at its previously unknown shortcomings.

There is little doubt that this type of transport is needed to improve upon, e.g., after many disappointing phase III trials of antibody-based medicines against Alzheimer's disease (listed in (Johnsen et al., 2019b)). Therefore, the TfR-mediated nanoparticle transport pathway remains an important avenue for therapeutics, as illustrated by a large number of preclinical evaluations and ongoing clinical trials (reviewed in (Johnsen et al., 2019b))

All in all, it is worthwhile to explore this avenue for therapeutics, especially with respect to understanding the transport directly, and not only on the level of tissue homogenates or downstream therapeutic efficacy. One striking example is an ongoing debate on the fundamental aspect of nanoparticle drug delivery, i.e., whether nanoparticle transcytosis is actually possible at all [see, e.g. (Freskgard and Ulrich, 2017)].

We have now highlighted this notion in the Discussion [Page 16, Line 9]

Rev1_Comment #3b. Most approaches for delivering therapeutic molecules across the BBB tend to target the capillaries. This is because capillaries are by far the most abundant vascular segment. By contrast, veins/venules are the least abundant.

Capillaries are the most abundant vascular segment, and we do not disagree with this fundamental feature of brain anatomy. The nanoparticles used in our study target both capillaries and venules, consistent with the expression pattern of the TfR (Kalucka et al., 2020; Sabbagh et al., 2018; Vanlandewijck et al., 2018). However, we indicate that the abundance of capillaries in the brain, further augmented by the highest binding of nanoparticles to this vessel segment, does not translate to an

efficient route of nanoparticle delivery to the brain. Conversely, a less abundant vessel segment, i.e., post-capillary venules, mediates nanoparticle brain delivery better than capillaries.

This may call for a reconsideration of using liposome nanoparticles as optimal drug vehicles via capillaries and instead target venules more specifically.

Rev1_Comment #4. Their data also suggests that few particles accumulate further into the parenchyma and are instead stuck in the perivascular space even after several days. How does that affect therapeutic potential?

It can be speculated that high retention in perivascular spaces may affect the pharmacokinetics of drug delivery or even cause adverse effects, but conversely, depending on the therapeutic cargo, it may be used for advantage in drug delivery as well. By retention in perivascular spaces and prolonged release of the therapeutics in proximity to mural cells, the nanoparticle-based therapies may be more efficient in the treatment of brain pathologies linked with the dysfunction of the neurovascular unit. These aspects would have to be assessed in a separate series of screening studies in various disease models with measurable functional readout and with distinct types of therapeutic cargo.

Further studies could also address the pharmacokinetics of payload release from nanoparticles by distinctively labeling the nanoparticle lipid membrane and the hydrophilic cargo within the aqueous nanoparticle core. This would allow to simultaneously monitor the fate of the carrier and the cargo in real-time in the brain in vivo. Of note, we are currently developing an imaging approach that is focused precisely on this aspect of nanoparticle drug delivery (Response Fig.1).

IMAGE REDACTED

Regardless, our current data provide very important information for drug nanocarrier design because, in contrast to the therapeutic payload in the brain, the fate of the nanoparticle carrier (clearance, location, etc.) is seldom addressed.

Rev1_Comment #5a. Could this persistent accumulation lead to adverse effects, even? The authors only show data claiming no adverse effects during a two hour imaging window (figure S1). But in some of their previous work with the same nanoparticle/antibody preparation they actually found significant adverse effects(Johnsen et al., 2019a).

First, we would like to underline that there are fundamental differences between the liposomal formulations used in our previous study referred to by Reviewer #1 (Johnsen et al., 2019a) and in the current paper. Most relevant is the fact that in our previous study, the antibodies were attached to the liposome nanoparticle surface using the post-insertion technique (Moreira et al., 2002), whereas here, we utilized the post-functionalization approach (Bak et al., 2016; Kristensen et al., 2019).

The post-insertion technique is a flexible method for adding different amounts of targeting ligands to the liposome surface, but it comes with the issue that lipid-conjugated antibodies that are not integrated into the liposomes may elicit immune responses in the circulation. In our previous study, when the liposomes with post-inserted R17 antibodies were administered (Johnsen et al., 2019a), the animals' reaction mirrored the reaction of animals receiving free antibodies (Couch et al., 2013; Weber et al., 2018). These side effects could be attributed to the retained effector function of the antibody Fc domain (Couch et al., 2013).

Notably, this was not observed for the liposomes used herein, where the antibodies were post-functionalized to the liposome surface. This is consistent with our previous reports, showing a lack of observed adverse effects when gold nanoparticles (Johnsen et al., 2018) or liposomes (Johnsen et al., 2017) were formulated with the approach used in the current manuscript.

We have now added this information in Supplementary Discussion.

Rev1_Comment #5b. Given that therapeutic benefit would likely come from repeated or extended exposures, time points beyond 2 hours should be explored to address this idea of a lack of adverse effects.

Reviewer #1 concludes that two hours is too short for the evaluation of side effects. We agree with the essence of this notion, but it is not true that adverse effects were only explored during two hours. The brain and systemic physiological parameters recorded during the acute imaging sessions were recorded over the course of two hours (Supplementary Fig. 1e), but the animals were monitored for days after administration of the nanoparticles in chronic imaging experiments.

It is important to note that TfR-mediated adverse effects as those observed in our previous study for liposomes or by Genentech and Roche for free antibodies started no later than five minutes post-injection (Couch et al., 2013; Johnsen et al., 2019a; Weber et al., 2018). The animals appeared lethargic, with limb spasticity, or death, which was not observed here. In our awake chronic imaging experiments the animals exhibited normal behavior (e.g., running, exploring, appetite) during and in between imaging sessions, with no observable differences before (24 h prior) and after (30 min, 24h, 48h) administration of nanoparticles. In addition, no adverse effects were observed in animals treated with liposomes with encapsulated cisplatin in experiments from Manuscript Fig. 1d.

This was mentioned in the Methods section, and we have now detailed it with the information in Supplementary Discussion.

Overall, Reviewer #1 is correct that all of the factors above are relevant when aiming to progress drug delivery into clinical trials and to obtain a lasting therapeutic effect, with an extended treatment regimen of weekly or monthly dosing. However, the assessment of the functional outcomes and therapeutic benefits vs. adverse effects were outside of the scope of this study, which is focused on the spatio-temporal aspects of nanoparticle transport across the BBB in vivo.

Rev1_Comment #5c. Furthermore, in line with this, it is becoming increasingly clear that post-capillary venules and the perivascular spaces are immunologically active under homeostatic conditions. Thus, it would therefore be warranted to ensure that nanoparticle delivery across this vascular segments and accumulation into these spaces does not disrupt homeostatic immune cell transmigration, which could potentially yield devastating side effects.

We agree that the venules are the important site for immune cell entry (Engelhardt and Ransohoff, 2012; Engelhardt et al., 2017) and should not be compromised by nanoparticle administration. In acute settings, it was clear that nanoparticle administration did not abolish the ability of immune cells to undergo the multistep process of transmigration, i.e., rolling, adhesion, and spreading (Supplementary Fig. 1; Supplementary Movie 2), which is the normal, expected behavior for blood-borne immune cells at post-capillary venules (Engelhardt and Ransohoff, 2012; Engelhardt et al., 2017). Furthermore, there were no signs that nanoparticles impaired diapedesis, as immune cells that contained nanoparticles could traverse across the BBB into perivascular spaces (Supplementary Movie 10). Regardless of acute or chronic imaging, we observed neither stalling nor accumulation of nanoparticle-rich immune cells getting arrested in perivascular spaces.

With regard to other aspects of nanoparticle-immune system interactions, earlier studies using a similar system of TfR-targeted immunoliposomes reported no side effects, e.g., neuroinflammation, during chronic weekly administrations (Zhang et al., 2003). Here, the nanoparticles were coated with poly(ethylene glycol; PEG) to reduce nanoparticle recognition by the immune system (Dobrovolskaia and McNeil, 2007; Moghimi, 2002). This may result in increased nanoparticle clearance due to the production of PEG-specific antibodies over subsequent injections, which was suggested to potentially change the pharmacokinetic profile, but without compromising immune system function (Dobrovolskaia and McNeil, 2007). In addition, we designed the nanoparticles to have near-neutral zeta potential (-10 to -7 mV)(Supplementary Table 1). This is of advantage, as a high negative or positive charge of nanoparticles may promote the formation of protein corona that interferes with nanoparticle targeting (Lundqvist et al., 2008); and increase nanoparticle exposure to the immune system, facilitating their uptake by phagocytes (Nakanishi et al., 1999; Xiao et al., 2011). Lastly, nanoparticle formulations used

herein do not also alter the TEER values in vitro, indicating a retained function of the paracellular barrier, as shown in our previous study (Johnsen et al., 2017).

This information has now been added to Supplementary Discussion.

We agree that when attempting the clinical translation of new drug delivery strategies, it is crucial to explore the long-term effects of transporting the nanoparticles into the brain. However, this study was not designed to assess the immune cell interactions, which would require the generation of multiple fluorescent reporter transgenic mice compatible with nanoparticle imaging to ascertain possible changes in the spatio-temporal pattern of immune cell entry to the brain.

Although we do not wish to argue that our data is an ultimate proof for the total absence of effects on immune cell entry, weighing in all available information, we respectfully disagree that there is a strong indicator regarding the nanoparticle transport process that warrants immune cell assessment to support our findings.

Rev1_Comment #6a. Claims based on two photon imaging would benefit from validation by histology. The authors rightly point out that tissue processing for histology can affect measurements of luminal bound nanoparticles. However, perfusion with fixative should effectively preserve any particles that are truly endocytosed and have no effect on their distribution (nor on the distribution of particles that have transited across the BBB completely). The live imaging is very important for this paper, but follow up by histology would provide the following:

- a) Superior spatial resolution. This would allow for more rigorous discrimination between particles that are truly inside versus outside endothelial cells.

We agree that the histological assessment in theory could improve nanoparticle localization estimates in the vicinity of the endothelium. In practice, however, perivascular spaces are nearly lost upon brain fixation. Recent work from Prof. Nedergaard's in *Nat. Commun.* (Mestre et al., 2018) demonstrated that the perivascular spaces undergo a 10-fold (!) decrease in volume upon fixation (see Response Fig. 2). This shrinkage leads to the displacement of perivascular space-resident nanoparticles and small fluorescent traces, forcing them back into vessel structure, including the basal lamina (Mestre et al., 2020). In addition, the loss of perivascular spaces leads to the efflux of macromolecules towards the pial surface of the brain (Mestre et al., 2020; Mestre et al., 2018). For these reasons, we did not expect brain fixation followed by immunohistochemistry to allow for good discrimination between particles inside and outside the endothelial cells.

IMAGE REDACTED

However, for the Reviewer's interest, we performed series of immunostaining assessments in mouse brains at 4 h and 24 h after nanoparticle administration. Following paraformaldehyde (PFA, 4%) perfusion-fixation, we observed a collapse of perivascular spaces with nanoparticles being relocated towards endothelial cell membranes. Moreover, although BECs retained their general morphology, the nanoparticle structure was compromised, likely due to perfusion-fixation combined with membrane permeabilization (perforation of lipid membranes) to enable antibody penetration for immunohistochemistry (See Response Fig. 3).

Response Fig. 3. Perfusion-fixation leads to relocation and destruction of nanoparticles. **a** Perfusion-fixation preserves the general *in vivo* morphology of endothelium for both, arterioles and venules. **b-c** However, it leads to collapse of perivascular space, and destruction of liposome nanoparticles (demonstrated by the presence of diffuse signal). **c** vessel orthographic projection along the dashed line. Round image insets denote the imaging plane in relation to a vessel perimeter. 'zs' insets indicate Z-stack maximum intensity projections.

Post-fixation artifacts effectively rendered further super-resolution imaging of nanoparticle location unreliable and exemplified the necessity of studying nanoparticle dynamics and distribution at the BBB interface in an intact living brain.

We have now added this information to the Results section [Page 7, Line 16] and [Page 11, Line 1], and the Response Fig. 3 panels were incorporated into Supplementary Fig. 3

Rev1_Comment #6b.

- b)** Greater understanding of molecular differences. For instance, transitions between capillaries and veins are not clearly identifiable by morphology and these areas could be potentially interesting. Indeed, a panel of carefully identified markers used in combination would likely be necessary to rigorously identify specific vascular segments, a strategy used in this study.

We agree that immunohistochemistry (IHC) might provide a more straightforward vessel classification; however, there are no established clear-cut molecular markers of transition from capillaries to post-capillary venules. For instance, leukocyte adhesion molecules are expressed in both capillaries and post-capillary venules and are suitable to distinguish only large venules from the capillary segment (Owens et al., 2008). Defining the proper molecular markers is further complicated by different transcriptome cluster classification criteria used in single-cell mRNA analyses of the brain endothelium, where the post-capillary segment is either not defined (Sabbagh et al., 2018; Vanlandewijck et al., 2018) or a practical term of, e.g., "capillary-venous" segment is coined, but without an anatomical foothold (Kalucka et al., 2020).

Here, similarly to most two-photon imaging assessments tracing vessel connectivity, we relied on vessel diameter and branching classification criteria. These criteria were based on the following:

- 1) *Two-photon imaging in vivo in mice reveals that capillary lumen diameters range on average from 10 to 5 μm (Cai et al., 2018; Hartmann et al., 2021; Khennouf et al., 2018;*

- Kutuzov et al., 2018). Importantly, this range reflects a gradual decrease in diameter along the cortical microvascular path, from the ~10 μm capillaries that stem from precapillary arterioles to terminal (~6th) branching orders of capillaries which do not exceed 5 μm diameter (Hartmann et al., 2021).
- 2) Following that, microvessels converge. Studies reconstructing brain vascular topology from mouse brain classify coalesced vessels with a diameter ~6-7 μm as the post-capillary venules (Cruz Hernandez et al., 2019; Smith et al., 2019). This division is based on the presence of loops in the capillary network, which are absent in post-capillary venules (Cruz Hernandez et al., 2019). This threshold is also used in functional assessments of the brain microvasculature, e.g., vascular oxygen partial pressure (PO_2) mapping in vivo, where ~7 μm microvessels are not considered to be a part of the capillary segment, but post-capillary venules (Sakadzic et al., 2014).
 - 3) The distinct morphology of mural cells supports vessel diameter-based division criteria. Capillaries at the venular end are covered by pericytes with a large cell body and numerous branches, whereas functionally and morphologically distinct 'mesh-like' (a.k.a. 'stellate' or 'post-capillary') pericytes reside at post-capillary venules (Hartmann et al., 2015; Uemura et al., 2020). The mesh-like pericytes occupy vessels with diameters no smaller than ~6-7 μm (Berthiaume et al., 2018; Hartmann et al., 2015), which is consistent with the chosen vessel diameter threshold for post-capillary venules.
 - 4) In addition to diameter and mural cells morphological differences, the lack of perivascular space at capillaries further distinguishes them from post-capillary venules. It is acknowledged that perivascular space is absent for vessels with a diameter of 6 μm or less, in contrast to the presence of perivascular space in post-capillary venules, which characterizes vessels with a diameter larger than 6 μm (Owens et al., 2008).

Overall, the data above converge on the definition of post-capillary venules as microvessels with a diameter larger than 6 μm and coalescing from two parent vessels located upstream of the blood flow.

We have now added this information to the Supplementary Discussion.

Rev1_Comment #6c.

- c) Broader sampling throughout other areas of the brain, including various clinically interesting areas not in the somatosensory cortex. Specifically, if nanoparticle delivery is primarily effective around only a subset of vessel segments it would be interesting to understand how those are distributed in the brain.

Two-photon laser scanning microscopy allows the assessment of naïve brain structure, but like every imaging approach, it has limitations. The most relevant one is the imaging depth, which allowed us to sample brain cortex up to ~250 μm below pia surface with a satisfactory resolution, but not deeper brain regions due to photon absorbance and scattering. In vivo imaging of, e.g., the hippocampus can be achieved by partial removal of the cortex and insertion of micro-optics lenses (Barretto et al., 2011; Jung et al., 2004; Levene et al., 2004), but the implantation is very invasive. Consequently, damaged brain microvasculature would create a massive number of extravasation spots obscuring the results.

However, we have performed a series of ex vivo immunohistochemistry experiments on tissue sections spanning the entire brain. Our data show that nanoparticles associated to vessels with the brain vessels in all regions of the brain (Response Fig. 4). However, due to the impact imposed by PFA fixation on perivascular space and on the nanoparticles (as shown in Response Fig. 3) it was impossible to distinguish whether a given nanoparticle was transcytosed or not, or assess nanoparticle distribution in perivascular spaces.

We have now commented now on this in the manuscript results [Page 11, Line 1], and added additional information to Supplementary Fig. 3.

Response Fig. 4. Broad immunohistochemistry sampling throughout brain areas at 4 h and 24 h after nanoparticle injection into the bloodstream. a Robust presence of nanoparticles in vicinity to venules and capillaries is observed in all brain regions, with traces of diffuse signal in the brain parenchyma. **b-c** No nanoparticle presence at arterioles. Vessels were classified as venules or arterioles based on BEC morphological features (see Response Fig 3).

Rev1_Comment #6d. This would also alleviate any concerns that imaging in an area where there was just surgery performed may affect the distribution of nanoparticles.

Although the preparation of craniotomy and imaging procedures may potentially compromise the function of the microvasculature, we rigorously implement internal controls in our imaging experiments.

- a) *In acutely imaged mice, we did not observe a loss of BBB integrity which would result in unspecific extravasation of nanoparticles from the bloodstream to the brain, even during prolonged sessions (~2h) of continuous imaging (e.g., Manuscript Fig. 1f).*
- b) *A significant number of our experiments were performed with nanoparticles co-injected with blood-circulating soluble fluorescent tracers to delineate the vasculature (FITC-dextran or TRITC-dextran). There was no leakage of tracers into the brain parenchyma, which indicated a non-compromised BBB.*
- c) *Prior to chronic imaging experiments, we tested our protocol using stealth (non-targeted) nanoparticles (Sth-L-A550) injected into the bloodstream. Chronic window preparation and imaging caused no extravasation or unspecific transport of nanoparticles to the brain parenchyma, even after recurring imaging sessions (see Response Fig. 5).*

Response Fig. 5. No extravasation of stealth nanoparticles (Sth-L-Atto550) into the brain parenchyma indicates preserved BBB integrity during chronic imaging. Images are maximum intensity projected Z-stacks. Time is relative to the time of Sth-L-A550 injection. All images are represented in the same fluorescence intensity scale.

- d) *Our recent two-photon experiments in vivo showed that imaging procedures applied here do not disturb the BBB. In healthy operated mice, the blood-circulating fluorescent albumin is not present in perivascular macrophages that reside in perivascular areas, in contrast to pathological conditions when transcytosis is altered (disinhibited) (Janiurek et al., 2019).*
- e) *Lastly, across many previous works, we demonstrated that the cranial window microsurgery applied here preserves neuronal, astrocyte, pericyte, and smooth muscle morphology and intracellular Ca²⁺ signaling (both spontaneous and evoked calcium signals)(Fordsmann et al., 2019; Grubb et al., 2020; Khennouf et al., 2018; Kucharz and Lauritzen, 2018; Lind et al., 2013). Functionally, we observed no detrimental effects on neurovascular coupling, which indicates a non-disrupted neurovascular unit (Cai et al., 2018; Fordsmann et al., 2019; Hall et al., 2014; Kucharz and Lauritzen, 2018; Kucharz et al., 2016). These assessments were performed using a combination of two-photon imaging with various techniques: brain electrophysiological recordings (generation of local field potentials, vessel contractility, propagation of vascular responses, and cerebral metabolic rate of oxygen consumption), intrinsic optical imaging, and laser speckle imaging.*

We have now added Response Fig. 5 to the manuscript as Supplementary Fig. 4.

Minor points

Rev1_Comment #7. Flow through different vessel segments varies significantly (velocity, turbulence, etc). It would be interesting to include some discussion how flow might play into the distribution of binding they observe.

We thank the Reviewer for this very interesting remark. In the healthy brain, the blood flow is generally assumed to be laminar (Fullstone et al., 2015), with a small percentage (~0.4%) of capillaries exhibiting transiently interrupted blood flow (Cruz Hernandez et al., 2019). Here, we observed no preferential association of nanoparticles at branching points or at vessels with high tortuosity, and no increase in binding of nanoparticles when blood-borne cells were temporarily obstructing the nanoparticles flow (Response Fig. 6).

Response Fig. 6. Transient stalling of nanoparticle flow (R17-L-A550, red arrow, at 1-3 min) caused by a white cell does not lead to preferential nanoparticle binding to endothelium. The plasma flow (green arrow, FITC-dx) remained continuous.

Given that brain vasculature hemodynamics differs between arterioles, capillaries, and venules (Response Fig. 7), it is plausible that the slowest blood velocity in capillaries may increase the time of interaction of nanoparticles with TfR, thus increasing the probability of nanoparticle binding. Yet, even in the vessels of similar capillary diameter, we have observed striking differences in nanoparticle association density (Manuscript Fig. 4d).

Brain vasculature hemodynamics gradients		
Parameter	Gradient	Source
Pressure	arterioles >> capillaries > venules	(Sweeney et al. 2018)(Sweeney et al., 2018)
Flow	venules > arterioles >> capillaries	(Sweeney et al. 2018)(Sweeney et al., 2018)
Velocity	arterioles > venules >> capillaries	(Santisakultarm et al. 2012)(Santisakultarm et al., 2012); (Sweeney et al. 2018)(Sweeney et al., 2018)

Response Fig. 7. Vascular hemodynamics with respect to distinct vessel types in the brain. Compiled overview.

We conclude that although blood pressure, flow, and velocity might influence binding, they may not play a decisive role in nanoparticle distribution, which follows more closely the vascular gradient of TfR expression (Kalucka et al., 2020; Sabbagh et al., 2018; Vanlandewijck et al., 2018).

This comment has now been added to the manuscript Discussion [Page 13, Line 1].

Rev1_Comment #8. Is the MSD analysis presented in figure 5 from all vessel segments observed or specifically from veins?

This MSD data represented nanoparticles from all vessels. We have now re-analyzed the data, linking each nanoparticle trace with its respective vessel location (Response Fig. 8). The updated results show that the MSD_v(t) of the nanoparticles exhibited significant deviation from linearity regardless of the vessel type ($p < 0.0001$ Wald-Wolfowitz runs test; $n_{cap} = 380$; $n_{pcV} = 200$; $n_{pV} = 90$ nanoparticles; in 5 mice total). This indicated that the nanoparticle movement was inconsistent with diffusion in all vessel types. The directional component was present in capillaries and post-capillary venules, where the MSD_v(t) exceeded values predicted by normal diffusion (linear fit) but was not apparent in pial vessels that

Response Fig. 8. MSD analysis with regard to distinct vessel types. Straight lines are linear fit.

exhibited anomalous average MSDv(t) trace. Although the MSD analysis does not disclose the underlying biological background, it suggested that the internalized nanoparticles do not move randomly but via coordinated intracellular trafficking in capillaries and post-capillary venules.

The Manuscript Fig. 5f and results section have now been updated [Page 9, Line 18].

Rev1_Comment #9. At what time point from nanoparticle delivery is the subcellular distribution in 5j from?

The subcellular distribution data has been collected 3 h after injection of nanoparticles into the bloodstream. This is now clarified in Results section [Page 9, Line 31], and in the Fig. 5 legend.

Rev1_Comment #10. Numbers of mice and numbers of individual vessels should be indicated more clearly (in addition to numbers of particles). For instance, it is not clear how many mice were used for the experiment in figure 6.

The information was present in the results text, but we have now included it in all figures as well.

REVIEWER #2

(Remarks to the Author):

I find the work from Kucharz et al. exceptionally well executed and with findings that can have a potential impact on several communities, including vasculature biology, blood-brain barrier physiology, nanomedicine, and neurology.

The quality of the in vivo imaging is supreme and equally matched by elegant and thorough data analysis. Such efforts alone make the work suitable for publication in ANY journal including Nature Communications.

We would like to thank Reviewer #2 for his insightful comments and recognition of our study.

Rev2_Comment #1a. However, before I can recommend the work to be published, I feel the authors should address two critical aspects. Imaging depth: I don't have a clear sense of where most of the imaging is executed and perhaps a detailed explanation either in the form of more information in the method or by making the scheme in figure 1b larger indicating in which part of the brain where most of the imaging takes place.

The imaging has been performed in the brain cortex via a cranial window at coordinates 3 mm lateral, 0.5 mm posterior to bregma, which corresponds to the location of somatosensory cortex in mice [see, e.g., functional map in Supplementary. Fig. 1b; (Kucharz and Lauritzen, 2018)]. The information on the location (somatosensory cortex) was present in the methods section but has now also been added to manuscript results [e.g., Page 4, Line 22 (acute); Page 11, Line 8 (chronic)] and throughout figures legends. We could reliably access the fluorescence signal from distinct nanoparticles up to 300 μm below pia surface.

In addition, we have also improved Manuscript Fig. 1b to indicate better the location of the craniotomy.

Rev2_Comment #1b. It appears the most blood vessels are part of the pia mater (...)

We would like to point out that the pial vessels represented only a small fraction of the imaged vasculature. The imaging has been performed in 3D, i.e., in Z-stack or Z-stack over time (hyperstack) mode, (Response Fig. 9). The data from Z-stacks underwent dimensionality reduction, i.e., Z-stacks were presented as maximum intensity projections.

Response Fig. 9. The principle of hyperstack imaging.

For the Reviewer's convenience, we attach the depth-coded projection of the microvasculature from a Z-stack example (Response Fig. 10), which illustrates that most of the imaged vasculature was at least >25 μm beneath the pia surface and embedded in the brain tissue.

Response Fig. 10. Three-dimensional reconstruction of the brain cortical microvasculature *in vivo*. **a)** perspective view; **b)** orthogonal side view projection; **c)** Z-stack maximum intensity projection; **d)** Z-stack depth color-coded projection. The imaging depth scale is relative to the pia surface.

In addition, we have now marked all projected Z-stacks in the manuscript figures with a 'Z-stack' image inset.

Rev2_Comment #1c. (...) and hence while technically considered part of the brain vasculature, their location within the meninges make them potentially phenotypically different from the brain parenchyma vasculature. They are characterised by less belong to an anatomical area which is lower pericyte density, served by a functioning lymphatic system and is connected to the peripheral nervous system via extrinsic innervations. All of these are the characteristics of a non-cerebral vasculature. While I can see why the author focussed on more accessible pial vessels, I wonder how much information is accessible from parenchymal vasculature and their implication in the whole blood-brain barrier.

We fully agree with Prof. Battaglia and do not consider leptomeningeal (pial) arterioles and venules as part of the canonical BBB. In fact, their non-cerebral phenotype was the reason why in our datasets, we kept them separate from other vessel segments. In contrast, diving segments of arterioles, precapillary arterioles, post-capillary venules, and ascending venules, despite being surrounded by perivascular space, are well embedded in the brain parenchyma. Therefore, together with capillaries, we considered these segments as an integral part of the cerebral microvasculature.

Rev2_Comment #2. Tf affinity/avidity. I'm not sure why the authors chose to use a high-affinity antibody against TfR, which is even more augmented by the liposome multivalency. It is now well established that ligands with high affinity (Yu et al., 2011) or high avidity (Niewoehner et al., 2014) for the transferrin receptor (TfR) cannot cross the BBB but instead redirect receptors to lysosomes for degradation. Villaseñor et al. showed that such an effect is regulated at the endosomes sorting level (Villaseñor et al., 2017). More recently, we established a similar impact of avidity to LRP1 receptors in modulating transcytosis (Tian et al., 2020), as well as showing two distinct pathways of intracellular transport, one associated with the F-bar protein syndapin-2 that leads to fast transcytosis with no endosomal sorting, and one that leads to endothelial internalisation and associated with endosomes and lysosomes. The liposomes chosen here belong to the high avidity class. The data show very eloquently their ability to

enter endothelial cells and guide the TfRs toward lysosomes in agreement with the existing literature. All the Discussion about deficient transcytosis and its relation with the venules/capillary etc. becomes redundant.

While I would love to see a similar analysis repeated with nanoparticles whose avidity is adequately optimized for transcytosis, I can see that such a request involves a considerable amount of work. On the other hand, I repeat myself, and I think the methodology here presented is very relevant and warrants publication. Perhaps a rewriting of the Discussion with consequent moderation of the current conclusions to include the current state-of-the-art on transcytosis might be still sufficient to justify the presented work keeping it compelling. One example is the role of perivascular space, and the recent conclusion is supported only if the TfR targeting liposomes were able to go via transcytosis.

We agree with Prof. Battaglia that standard formulation of the nanoparticles used in our study may be suboptimal for drug delivery, but given the large variety of nanoparticle types and targeting moieties used by different groups, we chose to adhere to a canonical nanoparticle formulation. By choosing this model nanoparticle system, we could better relate our findings to the broad spectrum of existing work, reaching as far as to the 1990s, where analyses of TfR-targeted nanoparticle transcytosis and intracellular sorting were initially attempted (Huwylar et al., 1996).

We fully support the view that avidity and affinity of targeting moiety is a significant factor in receptor-mediated drug delivery to the brain (Johnsen et al., 2018; Wiley et al., 2013), and have previously shown using Genentech antibodies that reduced avidity promotes transport of gold nanoparticles across the BBB (Johnsen et al., 2018). Here, our choice of antibody number per liposome corresponded to previous work on modifying the avidity of TfR-targeted liposomes, e.g., the original findings from Prof. Partridge's lab (Huwylar et al., 1996), which exemplified the pattern of uptake as a function of avidity, similar to conclusions from Prof. Battaglia's LRP1-targeting study (Tian et al., 2020). With regard to affinity, low-affinity antibodies may reduce the uptake of the nanoparticle by endothelium but increase the chances of nanoparticle exocytosis (Freskgard and Urich, 2017; Yu et al., 2011). However, as we discussed in our 2019 review, there are counterarguments from researchers working with high-affinity antibodies that should be considered in a balanced discussion (Johnsen et al., 2019b).

However, we respectfully disagree that the discussion regarding the vascular zonation of the nanoparticle transport in capillaries versus venules is redundant because the nanoparticle formulation (affinity/avidity) was not optimized to transport more.

In the revised manuscript, we have now commented on the arguments above and the current state-of-the-art of TfR-based brain drug delivery [Page 16, Line 2], underscoring how findings on free antibodies may be applicable to nanoparticles (a thorough discussion of this aspect can also be found in (Freskgard and Urich, 2017)), including the impact of the recent work from Prof. Battaglia's group (Tian et al., 2020).

Minor comment:

Rev2_Comment #3. General: I find the manuscript quite challenging to read, perhaps too segmented, with little flow between one section to the other. The figure captions are not useful and too hermetic.

We aimed to sequentially describe the fate of an intravenously injected nanoparticle in vivo, from the moment of injection to the bloodstream, association, followed by uptake to vascular endothelium, intracellular trafficking dynamics, and subcellular distribution, to exocytosis with progression in the brain parenchyma, within minutes, and then, over days.

We have now made the changes throughout the manuscript to improve the flow between these sub-chapters, and we have revised all figure legends to be more descriptive.

Rev2_Comment #4 (Page 3, Line 5): what's the difference between low molecular weight hydrophobic compounds and lipophilic drugs? Are the latter part of the former? Perhaps this sentence needs some clarification.

We thank the Reviewer for pointing this out. We have now clarified the text by rephrasing:

“(…) Transcellular diffusion across BEC is possible, but restricted to low-molecular weight hydrophobic compounds, and most lipophilic drugs show negligible brain uptake because of (…)”

to

“(…) Diffusion of molecules across BECs is possible but restricted to low-molecular weight hydrophobic compounds. Most of them, however, including therapeutics, show negligible brain uptake due to rapid outward transport by efflux pumps to the bloodstream” [Page 3, Line 6]

Rev2_Comment #5 (Page 6, line 13): how do the authors ensure that the overlapping signal is not an aggregation or an exchange between the two liposome formulations? Let's remind ourselves that the components that make liposomes have a **CMC>0** and thus exchange via assembly/disassembly processes.

We agree with Prof. Battaglia that lipids may exchange between individual liposome nanoparticles, or between liposome and endogenous lipid nanoparticles in the blood. In fact, we have previously studied this phenomenon for a range of different lipid-anchored fluorophores (Munter et al., 2018) (Kristensen et al., 2019). This work demonstrated that DPPE-anchored Atto dyes have a fairly low dissociation rate from liposome nanoparticles in plasma. At the same time, the dyes have high quantum yield and are photostable, making them a good choice for tracking the liposome nanoparticles in our present study.

This information has now been added to the Methods section.

These observations are further reinforced by our imaging data, providing more information as to whether two different fluorophores with the same lipid anchor exchange between liposomes after injection. Following the extravasation of nanoparticles that circulated in the bloodstream for 3 hours, we detected no presence of nanoparticles that exhibited both Atto 550 and Atto 488 fluorescence (Manuscript Fig 1i). This supports the notion that no events of fusion or aggregation occurred, as this would result in the exchange of membrane-bound fluorophores and fluorescence signal from both, Atto488 and Atto550. The nanoparticles might transiently appear as merged, but only when neighboring in distances less than the imaging resolution of the microscope (Manuscript Fig. 2h). Lastly, our collateral experiments show that even pressure-injection of the mix of Atto 550- and Atto 488-tagged nanoparticles did not lead to the fusion of liposomes, and only distinctively labeled nanoparticles were present in the brain parenchyma (See, also Rev2_Comment #7).

Rev2_Comment #6 (Page 8, line 6): Can the authors explain the omission with more details?

Here, we analyzed vessels aligned with the (x, y) focal plane, i.e., oriented perpendicular to the imaging axis (z) (vessel 'a', Response Fig. 11). Tracing nanoparticles from penetrating vessels that traversed the imaging plane under a steep angle, e.g., ascending venules (vessel 'b', Response Fig. 11), would result in an underestimation of the movement measured along v axis.

Furthermore, the nanoparticles would only briefly appear in the imaging plane. In order to obtain longer time recordings, the imaging would have to be limited to nanoparticles that exhibited only small degree of displacement (i.e., end-position) to remain in the focal plane.

Response Fig. 11. Nanoparticles that exhibit the same movement at identical model vessels **a** and **b**. Right panels represent top view. The movement measured along long symmetry axis v (red arrow=projection) is underestimated at vessel **b** because of the vessel orientation towards the imaging plane (x, y) .

Rev2_Comment #7 (Page 9, line 8-11). Is it possible that this is due to the advection within the perivascular space (see the work from Nedergaard and Iliff on the glymphatic system).

This is a very interesting remark, and we cannot rule out this possibility. To determine whether the movement is consistent with diffusion or advection requires the imaging of the distribution of diffuse

signal over time from small molecular traces, e.g., as in our previous work (Kutuzov et al., 2018) and by Iliff and colleagues (Iliff et al., 2012); or concurrent tracing of the movement of a large population of nanoparticles, similarly to more recent work from Prof. Nedergaard's lab (Mestre et al., 2018). In contrast, here, because of the scarcity of transcytosis-mediated delivery, we were able to image only single instances of nanoparticle movement, which is insufficient to establish whether nanoparticle motion satisfies diffusion or advection movement criteria.

Nonetheless, the comparisons between our and Prof. Nedergaard's work should be made with caution because of key differences between our approach and Prof. Nedergaard's. Here we observed a natural progression of a nanoparticle to the brain following transcytosis (Manuscript Fig. 6b-c), compared to infusion of large amounts of tracers or particles into the CNS (Iliff et al., 2012; Mestre et al., 2018). Infusion of fluids may potentially create an intracranial pressure gradient that may drive the bulk flow of solutes or macromolecules. Second, Mestre and colleagues used a different type of particle (i.e., polystyrene microspheres) (Mestre et al., 2018), which may move differently in the perivascular space. Lastly, their imaging was focused on arterioles while we investigated nanoparticles entering the perivascular space in venules. Although both vessel types exhibit pulsatile movement (Mestre et al., 2018; Santisakultarm et al., 2012), their potential to mediate bulk flow may significantly differ.

Of note, for the Reviewer's interest, the only instance when we observed a clear presence of convective flow was when the liposome nanoparticles were pressure-injected into the brain parenchyma via a glass micropipette/microelectrode (see Response Fig. 12).

Response Fig. 12. The directional movement of R17-L-A550 and R17-L-A488 nanoparticles that were pressure-injected into the brain parenchyma. Circles outline randomly selected nanoparticles. The time is relative to the start of the recording, i.e. ~5 minutes after pressure-injection (Kucharz K.; unpublished).

Prof Giuseppe Battaglia
Catalan Institution for Research and Advanced Studies
Institute for Bioengineering of Catalonia - Barcelona Institute of Science and Technology
Department of Chemistry and Institute for the Physics of Living Systems -University College London

REVIEWER #3

The blood brain barrier (BBB) is critical for maintaining the health of the brain, but is a major impediment to delivering drugs and other substances in a targeted manner. In order to get the substance of interest across the BBB, nanoparticles with surface moieties that make use of the endogenous transport machinery (usually transferrin) have been developed. However, where in the vasculature this transport is taking place was not known, as most of the studies had looked histologically, or with low resolution techniques like MRI. In this manuscript, the authors use in vivo two-photon microscopy to visualize where in the vascular network transport takes place. They find while a large number of particles are

taken up in the capillary bed, the transport of the particles stall within the endothelial cells of capillaries and thus do not make it into the brain. They find that most of the trans-vascular transport takes place in the venules, which they hypothesize to be due the larger perivascular space around venules.

I think this is a very nice manuscript. In general, the data is very clear and convincing. The analyses are well done and there is a great deal of attention paid to the limitations of the optical resolution in resolving the nano particle positions. Although the paper is largely observational, I think it is an important advance to the field and will guide the development of more effective BBB-crossing drug delivery systems.

We thank Reviewer #3 for his/her insightful comments and appreciation of our work.

I have only minor comments, listed below.

Rev3_Comment #1. The cited reference for transferrin expression in the capillaries is nearly four decades old, and the technology to assay single-cell level expression differences has greatly improved since then. What do more recent measurements of single-cell gene expression (such as those from the Betsholtz lab and others) show with regards to the expression patterns of transferrin (and other proteins involved in transcytosis) in the cerebral vasculature?

We referred to the seminal article authored by Jefferies (Jefferies et al., 1984), because it provided the first evidence of TfR expression in the brain endothelium compared to other peripheral endothelial cells. This was subsequently confirmed by others, including one of our earlier studies (Moos et al., 1998) and by complementary data to single-cell transcriptomics (Vanlandewijck et al., 2018). Noteworthy, these reports illustrate TfR presence on the level of receptor protein, not mRNA transcript.

Recent single-cell transcriptome data analyses provided valuable insight showing the non-uniform TfR mRNA presence (TFRC transcript) along the microvascular tree (we compiled expression data in Response Fig. 13). These studies outlined the presence of TFRC transcript along the vascular tree, being highest in capillaries, then venules, with only a moderate presence in arterioles (Kalucka et al., 2020; Sabbagh et al., 2018; Vanlandewijck et al., 2018).

Vascular zonation of TFRC expression in the brain				
Arterioles	Capillaries	Venules		Literature sources with links to corresponding online databases
+	++++	+++		(Vanlandewijck et al. 2018)(Vanlandewijck et al., 2018) https://betsholtzlab.org/VascularSingleCells/datababase.html
+	++	+		(Sabbagh et al. 2018)(Sabbagh et al., 2018) https://marksabbagh.shinyapps.io/vectrdb/
-/+	+	+++ 'capillary venous'	+'large venules'	(Kalucka et al. 2020)(Kalucka et al., 2020) https://endotheliomics.shinyapps.io/ec_atlas

Response Fig. 13. Vascular zonation of TFRC expression in the brain endothelium (compiled information).

Noteworthy, recent data from Prof. Wyss-Coray's lab, although not focused on TfR expression profile, exemplified that RMT transcytosis-related genes are in general expressed highly in venules, moderately in capillaries, and with low expression in arterioles(Yang et al., 2020). This suggests that overall, venules and capillaries be more predisposed to mediate RMT than arterioles.

In relation to our study, we observed no binding of TfR-targeted nanoparticles to the arteriolar branch of the brain microvasculature (Manuscript Fig. 2). Intriguingly, closer inspection of mRNA expression profile and immunohistochemistry data from Prof. Betsholtz's lab reveals the presence of TFRC transcript in arterioles, but its product, i.e., the TfR receptor protein, appears to be absent in arterioles on the immunostaining images (Extended Figure 3 in (Vanlandewijck et al., 2018)). This is consistent with the absence of nanoparticle binding to arterioles presented herein. Another interesting possibility is that if the TFRC transcript is present in arterioles and is translated to the receptor protein, this pool of TfR receptors may be functionally excluded from the uptake of nanoparticles from the bloodstream.

Of note, our two-photon imaging data from unrelated work on free antibody RMT shows a similar pattern of uptake. We observed that TfR antibodies (RI7217 and 8D3) do not bind to endothelium in arterioles, in contrast to robust binding in capillaries and venules (Response Fig. 14).

IMAGE REDACTED

The updated references, along with the short commentary, have now been added to the manuscript Discussion section [Page 12, Line 16].

Rev3_Comment #2a. Why do capillaries have the machinery for uptake but not complete transcytosis? This is odd.

We agree with Reviewer #3 and do not concede that capillaries are devoid of transcytosis machinery. There is no doubt that biologically, receptor-mediated transcytosis is present in all vessel types, with capillaries being the most-studied vascular segment of the blood-to-brain transport.

However, a critical distinction needs to be made between receptor-mediated transcytosis of endogenous ligands and transcytosis of nanoparticles, which may not follow the same intracellular route. Given the fundamental differences between native ligands and nanoparticles, it is conceivable that nanoparticle transcytosis may exhibit different extravasation profiles than endogenous ligands. While most of the research effort in the field is focused on understanding the trafficking of native (endogenous) ligands, where knowledge is sparse, the mechanisms of nanoparticle transcytosis in the brain endothelium are practically unknown. This is well-reflected in the recent review by the group of Ludovic Collin at Roche (Villasenor et al., 2019); and in a 2014 review by the group of Joan Abbott, where mechanisms on transcytosis at the BBB are referenced from studies that did not investigate the brain (Preston et al., 2014). A more recent review provided a more thorough summary of all existing evidence on the various steps of transcytosis in the brain endothelium, underscoring how little is established, especially for exocytosis (Toth et al., 2020).

In addition, it is important to stress that claims on the TfR ability to transcytose transferrin (its natural ligand) could be regarded as controversial by researchers studying iron uptake into the brain (Duck et al., 2017). The results from state-of-the-art models for this process are consistent with the dissociation of the iron atom during endosomal sorting and acidification from transferrin with subsequent iron efflux into the brain parenchyma via different transporters, including ferroportin, rather than transcytosis (Burkhardt et al., 2016; Duck et al., 2017; Skjorringe et al., 2015).

Rev3_Comment #2b. Is this a consequence of the size of the particles, and smaller particles might be transcytoses at a higher rate?

Liposome nanoparticle formulations tested for drug delivery range between 50-275 nm (Zylberberg and Matosevic, 2016). Here, we decided to adhere to the general standard of nanoparticle formulation tested in preclinical and clinical trials (Barenholz, 2012). The size is known to affect, e.g., the nanoparticle uptake in a tissue-dependent manner (Hatakeyama et al., 2004), and it is likely that it may affect intracellular sorting during transcytosis, and subsequent migration and distribution in the brain parenchyma. However, the sheer size may not be the sole determinant of the efficacy of nanoparticle delivery, as other parameters that go along with the changes in size may be of importance, e.g., nanoparticle receptor-targeting avidity (Johnsen et al., 2018), or, e.g., nanoparticle rigidity which is related to its lipid composition (e.g., saturated vs. non-saturated lipids ratio)(Hu et al., 2017).

These comments have now been added to the manuscript Discussion section [Page 15, Line 10].

Noteworthy, our methodological approach developed here may be an excellent platform to fine-tune liposome nanoparticle formulations, including size, for increased efficacy of drug delivery. We envisage that analogically to the approach presented in Manuscript Fig. 2g, the follow-up screening studies could focus on a head-to-head comparison between the distinctively labeled liposomes of different sizes and lipid compositions, or smaller lipid-based nanoparticles, e.g., micelles; or even distinct targeting moieties within the same brain in vivo.

Rev3_Comment #3. The larger perivascular space around venules seems to be a plausible explanation for the observed results, but in fairness, the authors do not conclusively rule out other hypotheses. Laser extravasation may produce different opening magnitudes and flows of fluid out of venules and capillaries, so this experiment is suggestive, but not conclusive. I think in light of this, the conclusion that the perivascular space difference could be toned down a bit.

The presence of perivascular space represents a plausible explanation of the contrasting extravasation profiles between venules and capillaries, consistent with the preferred route for nanoparticle progression in the brain observed by others (Mestre et al., 2018). However, we revised the Discussion to consider other factors that could potentially underlie the vascular zonation of nanoparticle entry to the brain.

These considerations have now been added to the Discussion section [Page 13, Line 17].

REFERENCES:

- Abbott, N.J., Patabendige, A.A., Dolman, D.E., Yusof, S.R., and Begley, D.J. (2010). Structure and function of the blood-brain barrier. *Neurobiol Dis* 37, 13-25.
- Apodaca, G., Katz, L.A., and Mostov, K.E. (1994). Receptor-mediated transcytosis of IgA in MDCK cells is via apical recycling endosomes. *J Cell Biol* 125, 67-86.
- Armulik, A., Genove, G., Mae, M., Nisancioglu, M.H., Wallgard, E., Niaudet, C., He, L., Norlin, J., Lindblom, P., Strittmatter, K., et al. (2010). Pericytes regulate the blood-brain barrier. *Nature* 468, 557-561.
- Bak, M., Jolck, R.I., Eliassen, R., and Andresen, T.L. (2016). Affinity Induced Surface Functionalization of Liposomes Using Cu-Free Click Chemistry. *Bioconj Chem* 27, 1673-1680.
- Bakker, E.N., Bacskai, B.J., Arbel-Ornath, M., Aldea, R., Bedussi, B., Morris, A.W., Weller, R.O., and Carare, R.O. (2016). Lymphatic Clearance of the Brain: Perivascular, Paravascular and Significance for Neurodegenerative Diseases. *Cell Mol Neurobiol* 36, 181-194.
- Barenholz, Y. (2012). Doxil(R)--the first FDA-approved nano-drug: lessons learned. *J Control Release* 160, 117-134.
- Barretto, R.P., Ko, T.H., Jung, J.C., Wang, T.J., Capps, G., Waters, A.C., Ziv, Y., Attardo, A., Recht, L., and Schnitzer, M.J. (2011). Time-lapse imaging of disease progression in deep brain areas using fluorescence microendoscopy. *Nature medicine* 17, 223-228.
- Berthiaume, A.A., Hartmann, D.A., Majesky, M.W., Bhat, N.R., and Shih, A.Y. (2018). Pericyte Structural Remodeling in Cerebrovascular Health and Homeostasis. *Front Aging Neurosci* 10, 210.
- Bien-Ly, N., Yu, Y.J., Bumbaca, D., Elstrott, J., Boswell, C.A., Zhang, Y., Luk, W., Lu, Y., Dennis, M.S., Weimer, R.M., et al. (2014). Transferrin receptor (TfR) trafficking determines brain uptake of TfR antibody affinity variants. *J Exp Med* 211, 233-244.
- Burkhart, A., Skjorringe, T., Johnsen, K.B., Siupka, P., Thomsen, L.B., Nielsen, M.S., Thomsen, L.L., and Moos, T. (2016). Expression of Iron-Related Proteins at the Neurovascular Unit Supports Reduction and Reoxidation of Iron for Transport Through the Blood-Brain Barrier. *Mol Neurobiol* 53, 7237-7253.
- Cabezón, I., Manich, G., Martín-Venegas, R., Camins, A., Pelegri, C., and Vilaplana, J. (2015). Trafficking of Gold Nanoparticles Coated with the 8D3 Anti-Transferrin Receptor Antibody at the Mouse Blood-Brain Barrier. *Mol Pharm* 12, 4137-4145.

Cai, C., Fordsmann, J.C., Jensen, S.H., Gesslein, B., Lonstrup, M., Hald, B.O., Zambach, S.A., Brodin, B., and Lauritzen, M.J. (2018). Stimulation-induced increases in cerebral blood flow and local capillary vasoconstriction depend on conducted vascular responses. *Proc Natl Acad Sci U S A* 115, E5796-E5804.

Couch, J.A., Yu, Y.J., Zhang, Y., Tarrant, J.M., Fujii, R.N., Meilandt, W.J., Solanoy, H., Tong, R.K., Hoyte, K., Luk, W., *et al.* (2013). Addressing safety liabilities of TfR bispecific antibodies that cross the blood-brain barrier. *Sci Transl Med* 5, 183ra157, 181-112.

Cruz Hernandez, J.C., Bracko, O., Kersbergen, C.J., Muse, V., Haft-Javaherian, M., Berg, M., Park, L., Vinarcsik, L.K., Ivasyk, I., Rivera, D.A., *et al.* (2019). Neutrophil adhesion in brain capillaries reduces cortical blood flow and impairs memory function in Alzheimer's disease mouse models. *Nat Neurosci* 22, 413-420.

Daneman, R., and Prat, A. (2015). The blood-brain barrier. *Cold Spring Harb Perspect Biol* 7, a020412.

De Bock, M., Van Haver, V., Vandenbroucke, R.E., Decrock, E., Wang, N., and Leybaert, L. (2016). Into rather unexplored terrain-transcellular transport across the blood-brain barrier. *Glia* 64, 1097-1123.

Dobrovolskaia, M.A., and McNeil, S.E. (2007). Immunological properties of engineered nanomaterials. *Nat Nanotechnol* 2, 469-478.

Duck, K.A., Simpson, I.A., and Connor, J.R. (2017). Regulatory mechanisms for iron transport across the blood-brain barrier. *Biochem Biophys Res Commun* 494, 70-75.

Engelhardt, B., and Ransohoff, R.M. (2012). Capture, crawl, cross: the T cell code to breach the blood-brain barriers. *Trends Immunol* 33, 579-589.

Engelhardt, B., Vajkoczy, P., and Weller, R.O. (2017). The movers and shapers in immune privilege of the CNS. *Nat Immunol* 18, 123-131.

Fordsmann, J.C., Murmu, R.P., Cai, C., Brazhe, A., Thomsen, K.J., Zambach, S.A., Lonstrup, M., Lind, B.L., and Lauritzen, M. (2019). Spontaneous astrocytic Ca(2+) activity abounds in electrically suppressed ischemic penumbra of aged mice. *Glia* 67, 37-52.

Freskgard, P.O., and Urich, E. (2017). Antibody therapies in CNS diseases. *Neuropharmacology* 120, 38-55.

Fullstone, G., Wood, J., Holcombe, M., and Battaglia, G. (2015). Modelling the Transport of Nanoparticles under Blood Flow using an Agent-based Approach. *Sci Rep* 5, 10649.

Grubb, S., Cai, C., Hald, B.O., Khennouf, L., Murmu, R.P., Jensen, A.G.K., Fordsmann, J., Zambach, S., and Lauritzen, M. (2020). Precapillary sphincters maintain perfusion in the cerebral cortex. *Nat Commun* 11, 395.

Hadaczek, P., Yamashita, Y., Mirek, H., Tamas, L., Bohn, M.C., Noble, C., Park, J.W., and Bankiewicz, K. (2006). The "perivascular pump" driven by arterial pulsation is a powerful mechanism for the distribution of therapeutic molecules within the brain. *Mol Ther* 14, 69-78.

Hall, C.N., Reynell, C., Gesslein, B., Hamilton, N.B., Mishra, A., Sutherland, B.A., O'Farrell, F.M., Buchan, A.M., Lauritzen, M., and Attwell, D. (2014). Capillary pericytes regulate cerebral blood flow in health and disease. *Nature* 508, 55-60.

Hartmann, D.A., Berthiaume, A.A., Grant, R.I., Harrill, S.A., Koski, T., Tieu, T., McDowell, K.P., Faino, A.V., Kelly, A.L., and Shih, A.Y. (2021). Brain capillary pericytes exert a substantial but slow influence on blood flow. *Nat Neurosci*.

Hartmann, D.A., Underly, R.G., Grant, R.I., Watson, A.N., Lindner, V., and Shih, A.Y. (2015). Pericyte structure and distribution in the cerebral cortex revealed by high-resolution imaging of transgenic mice. *Neurophotonics* 2, 041402.

Hatakeyama, H., Akita, H., Maruyama, K., Suhara, T., and Harashima, H. (2004). Factors governing the in vivo tissue uptake of transferrin-coupled polyethylene glycol liposomes in vivo. *Int J Pharm* 281, 25-33.

Hellinger, E., Veszelka, S., Toth, A.E., Walter, F., Kittel, A., Bakk, M.L., Tihanyi, K., Hada, V., Nakagawa, S., Duy, T.D., *et al.* (2012). Comparison of brain capillary endothelial cell-based and epithelial (MDCK-MDR1, Caco-2, and VB-Caco-2) cell-based surrogate blood-brain barrier penetration models. *Eur J Pharm Biopharm* 82, 340-351.

Hu, Y., Rip, J., Gaillard, P.J., de Lange, E.C.M., and Hammarlund-Udenaes, M. (2017). The Impact of Liposomal Formulations on the Release and Brain Delivery of Methotrexate: An In Vivo Microdialysis Study. *J Pharm Sci* 106, 2606-2613.

Huotari, J., and Helenius, A. (2011). Endosome maturation. *Embo J* 30, 3481-3500.

Huwlyer, J., Wu, D., and Pardridge, W.M. (1996). Brain drug delivery of small molecules using immunoliposomes. *Proc Natl Acad Sci U S A* 93, 14164-14169.

Illiff, J.J., Wang, M., Liao, Y., Plogg, B.A., Peng, W., Gundersen, G.A., Benveniste, H., Vates, G.E., Deane, R., Goldman, S.A., *et al.* (2012). A paravascular pathway facilitates CSF flow through the brain parenchyma and the clearance of interstitial solutes, including amyloid beta. *Sci Transl Med* 4, 147ra111.

Janiurek, M., Soylyu-Kucharz, R., Christoffersen, C., Kucharz, K., and Lauritzen, M. (2019). Apolipoprotein M-bound sphingosine-1-phosphate regulates blood-brain barrier paracellular permeability and transcytosis. *Elife* 8.

Jefferies, W.A., Brandon, M.R., Hunt, S.V., Williams, A.F., Gatter, K.C., and Mason, D.Y. (1984). Transferrin receptor on endothelium of brain capillaries. *Nature* 312, 162-163.

Johnsen, K.B., Bak, M., Kempen, P.J., Melander, F., Burkhart, A., Thomsen, M.S., Nielsen, M.S., Moos, T., and Andresen, T.L. (2018). Antibody affinity and valency impact brain uptake of transferrin receptor-targeted gold nanoparticles. *Theranostics* 8, 3416-3436.

Johnsen, K.B., Bak, M., Melander, F., Thomsen, M.S., Burkhart, A., Kempen, P.J., Andresen, T.L., and Moos, T. (2019a). Modulating the antibody density changes the uptake and transport at the blood-brain barrier of both transferrin receptor-targeted gold nanoparticles and liposomal cargo. *J Control Release* 295, 237-249.

Johnsen, K.B., Burkhart, A., Melander, F., Kempen, P.J., Vejlebo, J.B., Siupka, P., Nielsen, M.S., Andresen, T.L., and Moos, T. (2017). Targeting transferrin receptors at the blood-brain barrier improves the uptake of immunoliposomes and subsequent cargo transport into the brain parenchyma. *Sci Rep* 7, 10396.

Johnsen, K.B., Burkhart, A., Thomsen, L.B., Andresen, T.L., and Moos, T. (2019b). Targeting the transferrin receptor for brain drug delivery. *Prog Neurobiol* 181, 101665.

Johnsen, K.B., and Moos, T. (2016). Revisiting nanoparticle technology for blood-brain barrier transport: Unfolding at the endothelial gate improves the fate of transferrin receptor-targeted liposomes. *J Control Release* 222, 32-46.

Jung, J.C., Mehta, A.D., Aksay, E., Stepnoski, R., and Schnitzer, M.J. (2004). In vivo mammalian brain imaging using one- and two-photon fluorescence microendoscopy. *Journal of neurophysiology* 92, 3121-3133.

Kalucka, J., de Rooij, L., Goveia, J., Rohlenova, K., Dumas, S.J., Meta, E., Conchinha, N.V., Taverna, F., Teuwen, L.A., Veys, K., *et al.* (2020). Single-Cell Transcriptome Atlas of Murine Endothelial Cells. *Cell* 180, 764-779 e720.

Khenouf, L., Gesslein, B., Brazhe, A., Oceau, J.C., Kutuzov, N., Khakh, B.S., and Lauritzen, M. (2018). Active role of capillary pericytes during stimulation-induced activity and spreading depolarization. *Brain* 141, 2032-2046.

Kristensen, K., Engel, T.B., Stensballe, A., Simonsen, J.B., and Andresen, T.L. (2019). The hard protein corona of stealth liposomes is sparse. *J Control Release* 307, 1-15.

Kucharz, K., and Lauritzen, M. (2018). CaMKII-dependent endoplasmic reticulum fission by whisker stimulation and during cortical spreading depolarization. *Brain* 141, 1049-1062.

Kucharz, K., Sondergaard Rasmussen, I., Bach, A., Stromgaard, K., and Lauritzen, M. (2016). PSD-95 uncoupling from NMDA receptors by Tat-N-dimer ameliorates neuronal depolarisation in cortical spreading depression. *J Cereb Blood Flow Metab.*

Kutuzov, N., Flyvbjerg, H., and Lauritzen, M. (2018). Contributions of the glycocalyx, endothelium, and extravascular compartment to the blood-brain barrier. *Proc Natl Acad Sci U S A* 115, E9429-E9438.

Levene, M.J., Dombeck, D.A., Kasischke, K.A., Molloy, R.P., and Webb, W.W. (2004). In vivo multiphoton microscopy of deep brain tissue. *Journal of neurophysiology* 91, 1908-1912.

Lind, B.L., Brazhe, A.R., Jessen, S.B., Tan, F.C., and Lauritzen, M.J. (2013). Rapid stimulus-evoked astrocyte Ca²⁺ elevations and hemodynamic responses in mouse somatosensory cortex in vivo. *Proc Natl Acad Sci U S A* 110, E4678-4687.

Lundqvist, M., Stigler, J., Elia, G., Lynch, I., Cedervall, T., and Dawson, K.A. (2008). Nanoparticle size and surface properties determine the protein corona with possible implications for biological impacts. *Proc Natl Acad Sci U S A* 105, 14265-14270.

Mestre, H., Mori, Y., and Nedergaard, M. (2020). The Brain's Glymphatic System: Current Controversies. *Trends Neurosci* 43, 458-466.

Mestre, H., Tithof, J., Du, T., Song, W., Peng, W., Sweeney, A.M., Olveda, G., Thomas, J.H., Nedergaard, M., and Kelley, D.H. (2018). Flow of cerebrospinal fluid is driven by arterial pulsations and is reduced in hypertension. *Nat Commun* 9, 4878.

Moghimi, S.M. (2002). Chemical camouflage of nanospheres with a poorly reactive surface: towards development of stealth and target-specific nanocarriers. *Biochimica et biophysica acta* 1590, 131-139.

Moos, T., Oates, P.S., and Morgan, E.H. (1998). Expression of the neuronal transferrin receptor is age dependent and susceptible to iron deficiency. *J Comp Neurol* 398, 420-430.

Moreira, J.N., Ishida, T., Gaspar, R., and Allen, T.M. (2002). Use of the post-insertion technique to insert peptide ligands into pre-formed stealth liposomes with retention of binding activity and cytotoxicity. *Pharm Res* 19, 265-269.

Muldoon, L.L., Pagel, M.A., Kroll, R.A., Roman-Goldstein, S., Jones, R.S., and Neuwelt, E.A. (1999). A physiological barrier distal to the anatomic blood-brain barrier in a model of transvascular delivery. *AJNR Am J Neuroradiol* 20, 217-222.

Munter, R., Kristensen, K., Pedersbaek, D., Larsen, J.B., Simonsen, J.B., and Andresen, T.L. (2018). Dissociation of fluorescently labeled lipids from liposomes in biological environments challenges the interpretation of uptake studies. *Nanoscale* 10, 22720-22724.

Nakanishi, T., Kunisawa, J., Hayashi, A., Tsutsumi, Y., Kubo, K., Nakagawa, S., Nakanishi, M., Tanaka, K., and Mayumi, T. (1999). Positively charged liposome functions as an efficient immunoadjuvant in inducing cell-mediated immune response to soluble proteins. *J Control Release* 61, 233-240.

Niewoehner, J., Bohrmann, B., Collin, L., Urich, E., Sade, H., Maier, P., Rueger, P., Stracke, J.O., Lau, W., Tissot, A.C., *et al.* (2014). Increased brain penetration and potency of a therapeutic antibody using a monovalent molecular shuttle. *Neuron* 81, 49-60.

Owens, T., Bechmann, I., and Engelhardt, B. (2008). Perivascular spaces and the two steps to neuroinflammation. *J Neuropathol Exp Neurol* 67, 1113-1121.

Preston, J.E., Joan Abbott, N., and Begley, D.J. (2014). Transcytosis of macromolecules at the blood-brain barrier. *Adv Pharmacol* 71, 147-163.

Sabbagh, M.F., Heng, J.S., Luo, C., Castanon, R.G., Nery, J.R., Rattner, A., Goff, L.A., Ecker, J.R., and Nathans, J. (2018). Transcriptional and epigenomic landscapes of CNS and non-CNS vascular endothelial cells. *Elife* 7.

Sabbagh, M.F., and Nathans, J. (2020). A genome-wide view of the de-differentiation of central nervous system endothelial cells in culture. *Elife* 9.

Sakadzic, S., Mandeville, E.T., Gagnon, L., Musacchia, J.J., Yaseen, M.A., Yucel, M.A., Lefebvre, J., Lesage, F., Dale, A.M., Eikermann-Haerter, K., *et al.* (2014). Large arteriolar component of oxygen delivery implies a safe margin of oxygen supply to cerebral tissue. *Nat Commun* 5, 5734.

Santisakultarm, T.P., Cornelius, N.R., Nishimura, N., Schafer, A.I., Silver, R.T., Doerschuk, P.C., Olbricht, W.L., and Schaffer, C.B. (2012). In vivo two-photon excited fluorescence microscopy reveals cardiac- and respiration-dependent pulsatile blood flow in cortical blood vessels in mice. *Am J Physiol Heart Circ Physiol* 302, H1367-1377.

Skjorringe, T., Burkhart, A., Johnsen, K.B., and Moos, T. (2015). Divalent metal transporter 1 (DMT1) in the brain: implications for a role in iron transport at the blood-brain barrier, and neuronal and glial pathology. *Front Mol Neurosci* 8, 19.

Smith, A.F., Doyeux, V., Berg, M., Peyrounette, M., Haft-Javaherian, M., Larue, A.E., Slater, J.H., Lauwers, F., Blinder, P., Tsai, P., *et al.* (2019). Brain Capillary Networks Across Species: A few Simple Organizational Requirements Are Sufficient to Reproduce Both Structure and Function. *Front Physiol* 10, 233.

Sweeney, P.W., Walker-Samuel, S., and Shipley, R.J. (2018). Insights into cerebral haemodynamics and oxygenation utilising in vivo mural cell imaging and mathematical modelling. *Sci Rep* 8, 1373.

Tian, X., Leite, D.M., Scarpa, E., Nyberg, S., Fullstone, G., Forth, J., Matias, D., Apriceno, A., Poma, A., Duro-Castano, A., *et al.* (2020). On the shuttling across the blood-brain barrier via tubule formation: Mechanism and cargo avidity bias. *Sci Adv* 6.

Toth, A.E., Holst, M.R., and Nielsen, M.S. (2020). Vesicular Transport Machinery in Brain Endothelial Cells: What We Know and What We Do not. *Curr Pharm Des* 26, 1405-1416.

Toth, A.E., Nielsen, S.S.E., Tomaka, W., Abbott, N.J., and Nielsen, M.S. (2019). The endo-lysosomal system of bEnd.3 and hCMEC/D3 brain endothelial cells. *Fluids Barriers CNS* 16, 14.

Toth, A.E., Siupka, P., TJ, P.A., Veno, S.T., Thomsen, L.B., Moos, T., Lohi, H.T., Madsen, P., Lykke-Hartmann, K., and Nielsen, M.S. (2018). The Endo-Lysosomal System of Brain Endothelial Cells Is Influenced by Astrocytes In Vitro. *Mol Neurobiol* 55, 8522-8537.

Uemura, M.T., Maki, T., Ihara, M., Lee, V.M.Y., and Trojanowski, J.Q. (2020). Brain Microvascular Pericytes in Vascular Cognitive Impairment and Dementia. *Front Aging Neurosci* 12, 80.

van Rooy, I., Mastrobattista, E., Storm, G., Hennink, W.E., and Schiffelers, R.M. (2011). Comparison of five different targeting ligands to enhance accumulation of liposomes into the brain. *J Control Release* 150, 30-36.

Vanlandewijck, M., He, L., Mae, M.A., Andrae, J., Ando, K., Del Gaudio, F., Nahar, K., Lebouvier, T., Lavina, B., Gouveia, L., *et al.* (2018). A molecular atlas of cell types and zonation in the brain vasculature. *Nature* 554, 475-480.

Villasenor, R., Lampe, J., Schwaninger, M., and Collin, L. (2019). Intracellular transport and regulation of transcytosis across the blood-brain barrier. *Cell Mol Life Sci* 76, 1081-1092.

Villasenor, R., Schilling, M., Sundaresan, J., Lutz, Y., and Collin, L. (2017). Sorting Tubules Regulate Blood-Brain Barrier Transcytosis. *Cell Rep* 21, 3256-3270.

Weber, F., Bohrmann, B., Niewoehner, J., Fischer, J.A.A., Rueger, P., Tiefenthaler, G., Moelleken, J., Bujotzek, A., Brady, K., Singer, T., *et al.* (2018). Brain Shuttle Antibody for Alzheimer's Disease with Attenuated Peripheral Effector Function due to an Inverted Binding Mode. *Cell Rep* 22, 149-162.

Wiley, D.T., Webster, P., Gale, A., and Davis, M.E. (2013). Transcytosis and brain uptake of transferrin-containing nanoparticles by tuning avidity to transferrin receptor. *Proc Natl Acad Sci U S A* 110, 8662-8667.

Xiao, K., Li, Y., Luo, J., Lee, J.S., Xiao, W., Gonik, A.M., Agarwal, R.G., and Lam, K.S. (2011). The effect of surface charge on in vivo biodistribution of PEG-oligocholic acid based micellar nanoparticles. *Biomaterials* 32, 3435-3446.

Yang, A.C., Stevens, M.Y., Chen, M.B., Lee, D.P., Stahli, D., Gate, D., Contrepolis, K., Chen, W., Iram, T., Zhang, L., *et al.* (2020). Physiological blood-brain transport is impaired with age by a shift in transcytosis. *Nature*.

Yu, Y.J., Zhang, Y., Kenrick, M., Hoyte, K., Luk, W., Lu, Y., Atwal, J., Elliott, J.M., Prabhu, S., Watts, R.J., and Dennis, M.S. (2011). Boosting brain uptake of a therapeutic antibody by reducing its affinity for a transcytosis target. *Sci Transl Med* 3, 84ra44.

Zhang, Y.F., Boado, R.J., and Pardridge, W.M. (2003). Absence of toxicity of chronic weekly intravenous gene therapy with pegylated immunoliposomes. *Pharm Res* 20, 1779-1785.

Zylberberg, C., and Matosevic, S. (2016). Pharmaceutical liposomal drug delivery: a review of new delivery systems and a look at the regulatory landscape. *Drug Deliv* 23, 3319-3329.

Reviewers' Comments:

Reviewer #1:

Remarks to the Author:

In this revision, Kucharz et al. do a good job clarifying the issues previously raised by the reviewers. Specifically the following points:

- The authors do a good job in this revision's discussion going over the possible alternative interpretations of their data
- Inclusion of IHC data is important to show that the observed associations are not just a product of imaging near the surface of the brain at the site of a craniotomy
- The revisions to the text (particularly discussion) do a good job clarifying semantic ambiguities, including definitions of vascular segments and discussion of trafficking routes across vessels
- The clarification regarding post-insertion versus post-functionalization modification of nanoparticles is also helpful

Only a couple of minor points:

- Figure 5 explicitly quantifies the subcellular distribution of nanoparticles in venous endothelial cells (vECs). It is stated that the perinuclear distribution observed in vECs is not observed in capillary endothelial cells (cECs). Although example images are provided in the figure, no quantification is provided. It would be helpful to present that data for cECs (same as fig 5j,k) to get a sense for how different the distributions really are.
- On line 453 the authors state they do not observe association of nanoparticles with neurons, which might be expected due to the presence of transferrin receptor on neurons. It is not clear if they are making this claim solely from their live imaging studies, but if they are, there are relatively few neuronal somata in layer 1 of cortex, which is where they are imaging. Therefore it is difficult to state a lack of association with neurons as a certainty.

Reviewer #2:

Remarks to the Author:

I'm very pleased with the changes and the manuscript has considerably improved. All my questions have been addressed and I'm more than happy to recommend the manuscript publication.

Reviewer #3:

Remarks to the Author:

All my comments have been addressed, it is a very nice result.

2nd Revision

REVIEWER #1 (Remarks to the Author)

Reviewer's Comments:

In this revision, Kucharz et al. do a good job clarifying the issues previously raised by the reviewers. Specifically, the following points:

- The authors do a good job in this revision's discussion going over the possible alternative interpretations of their data
- Inclusion of IHC data is important to show that the observed associations are not just a product of imaging near the surface of the brain at the site of a craniotomy
- The revisions to the text (particularly discussion) do a good job clarifying semantic ambiguities, including definitions of vascular segments and discussion of trafficking routes across vessels
- The clarification regarding post-insertion versus post-functionalization modification of nanoparticles is also helpful

We thank Reviewer #1 for the insightful comments, constructive criticism, and suggestions on text clarifications and control experiments, which helped improve the manuscript.

Only a couple of minor points:

Rev1_Comment #1. Figure 5 explicitly quantifies the subcellular distribution of nanoparticles in venous endothelial cells (vECs). It is stated that the perinuclear distribution observed in vECs is not observed in capillary endothelial cells (cECs). Although example images are provided in the figure, no quantification is provided. It would be helpful to present that data for cECs (same as fig 5j,k) to get a sense for how different the distributions really are.

We agree that a direct comparison of cECs and vECs would be informative. However, the quantifications from capillaries, when performed in the same manner as for larger venules, would not be reliable. This is due to the limited two-photon imaging resolution in z (=depth) dimension, small capillary diameter, and complex three-dimensional organization of cECs.

In order to compare cECs to vECs by measuring Euclidean distances of nanoparticles from the nucleus boundary, first, an assumption would need to be made that a capillary represents a straight cylindrical structure. This is not observed in vivo, and adjusting the structure geometry in image post-processing (e.g., by straightening) would alter the spatial properties of cell morphology and influence the local coordinate system (Response Fig. 15a; blue squares). Second, even if the perfectly cylindrical capillary were found, the endothelial cell that wraps around the vessel lumen would need to be transformed into a planar surface (Response Fig. 15b). However, due to the limitation of two-photon microscopy resolution, with Z point spread function reaching ~3-5 μm , this was not feasible, as the fine details of endothelial cell boundaries are obscured along Z-axis (Response Fig. 15c).

Response Fig. 15. **a.** Straightening of a capillary vessel along its long axis results in changes in the local coordinate system (blue squares). **b.** 3D transformation of hypothetical endothelial cell (green) into a planar structure. **c.** brain capillary microvessel in (x,y) imaging plane with a respective projection in the z (=depth) dimension along a demarked line. **d.** measurements of nanoparticle distances from nucleus boundary at cECs without 3D transformation (d4), may result in significant distance underestimation compared to an actual distribution (d1 vs d4).

To demonstrate the importance of this limitation, one can model a theoretical cEC with a homogeneous nanoparticle distribution (Response Fig. 15d, d1). If the planar projections of cECs were avoided, the quantifications would need to rely on non-planar representations of cECs (Response Fig. 15d; d4). This would lead to heterogeneous underestimation of the measured distances between the nanoparticles and the nuclear boundary (Response Fig. 15 d; red vs. yellow bars). Consequently, such analysis would not provide the actual nanoparticle distribution in the cell (Response Fig. 15 d1 vs. d4). Lastly, it is rare for the nucleus to be oriented in the middle of the vessel projection, as illustrated in Response Fig. 15b,d. In the vast majority of capillaries, the nucleus resides on the border of a capillary vessel (Response Fig. 15c), which could further increase the nucleus-nanoparticle measurement error.

The alternative approach might require serial sectioning and subsequent transmission electron microscopy (TEM), or serial block face/focused ion beam milling scanning electron microscopy (SBF-SEM and FIB-SEM, respectively) for 3D reconstruction, which could potentially work for cECs; however, the lipid nanoparticles are obliterated during the sample preparation process for EM. Furthermore, in contrast to capillaries, sample processing for EM is known to significantly alter the morphology of large arterioles and venules (by tissue shrinkage and non-uniform distortion of the endothelium in proximity to tissue-devoid areas, i.e., vessel lumen). Thus, comparisons between vECs and cECs would be very sensitive to fixation artifacts.

Taking all into account, we rephrased our notion:

From:

“(…) The nanoparticles localized over time to the perinuclear region of BECs in venules (Fig. 5g), but not in capillaries (Fig. 5h). We quantified the spatial distribution of nanoparticles 3 h post-injection in venules by measuring their distances from the geometric center of the nucleus (Fig. 5i).”

To:

“(…) The nanoparticles localized over time to the perinuclear region of BECs in venules (Fig. 5g), but this was not observed in capillaries (Fig. 5h). We quantified the spatial distribution of nanoparticles in venules 3 h post-injection by measuring their Euclidean distances from the geometric center of the nucleus (Fig. 5i). We refrained from measurements in capillaries because of systematic underestimation of nanoparticle distances from the nucleus due to high vessel curvature. (…)”

Rev1_Comment #2. On line 453 the authors state they do not observe association of nanoparticles with neurons, which might be expected due to the presence of transferrin receptor on neurons. It is not clear if they are making this claim solely from their live imaging studies, but if they are, there are relatively few neuronal somata in layer 1 of cortex, which is where they are imaging. Therefore it is difficult to state a lack of association with neurons as a certainty.

The presence of neuronal somata in layer 1 is sparse compared to deeper cortical layers, but some neuronal cell bodies can be observed at depths ~200 μm below the pia surface using two-photon microscopy [e.g., as in Figure 1 and Supplementary Fig. 4a in (Kucharz and Lauritzen, 2018)]. Our comment was based on visual inspections of Z-stacks from imaging experiments, where we detected no clustering of nanoparticles, suggestive of no binding to neuronal somata. However, we agree with Reviewer #1, that this observation does not prove the lack of nanoparticle interactions/binding to neurons.

We have now removed the following sentence from the Discussion section:

(...) In the murine brain, neurons are located on average within short (~15 μm) distances from brain capillaries, and although TfR may potentially act as an intraparenchymal target for neurons (Farias et al., 2015), we observed no cellular association patterns of the nanoparticles suggestive of binding to TfRs, which are enriched at the neuronal somata (Farias et al., 2015).

REVIEWER #2 (Remarks to the Author)

Reviewer's Comments:

I'm very pleased with the changes and the manuscript has considerably improved. All my questions have been addressed and I'm more than happy to recommend the manuscript publication.

We thank Reviewer #2 for the insightful comments and suggestions concerning data analysis, presentation, and discussion, which helped improve the quality of the manuscript.

REVIEWER #3 (Remarks to the Author)

Reviewer's Comments:

All my comments have been addressed, it is a very nice result.

We thank Reviewer #3 for the insightful comments that helped improve the manuscript, and intriguing questions that call for exciting follow-up studies.

REFERENCES

- Farias, G.G., Guardia, C.M., Britt, D.J., Guo, X., and Bonifacino, J.S. (2015). Sorting of Dendritic and Axonal Vesicles at the Pre-axonal Exclusion Zone. *Cell Rep* 13, 1221-1232.
- Kucharz, K., and Lauritzen, M. (2018). CaMKII-dependent endoplasmic reticulum fission by whisker stimulation and during cortical spreading depolarization. *Brain* 141, 1049-1062.